# LAGRANGIAN FLOW NETWORKS FOR CONSERVATION LAWS

**Fabricio Arend Torres, Marcello M. Negri, Marco Inversi, Jonathan Aellen & Volker Roth**
Department of Mathematics and Computer Science
University of Basel
{fabricio.arendtorres, marcellomassimo.negri,
 marco.inversi, jonathan.aellen, volker.roth}@unibas.ch

## ABSTRACT

We introduce *Lagrangian Flow Networks* (LFlows) for modeling fluid densities and velocities continuously in space and time. By construction, the proposed LFlows satisfy the continuity equation, a PDE describing mass conservation in its differential form. Our model is based on the insight that solutions to the continuity equation can be expressed as time-dependent density transformations via differentiable and invertible maps. This follows from classical theory of the existence and uniqueness of Lagrangian flows for smooth vector fields. Hence, we model fluid densities by transforming a base density with parameterized diffeomorphisms conditioned on time. The key benefit compared to methods relying on numerical ODE solvers or PINNs is that the analytic expression of the velocity is always consistent with changes in density. Furthermore, we require neither expensive numerical solvers, nor additional penalties to enforce the PDE. LFlows show higher predictive accuracy in density modeling tasks compared to competing models in 2D and 3D, while being computationally efficient. As a real-world application, we model bird migration based on sparse weather radar measurements.

## 1 INTRODUCTION

The development of physics-informed Machine Learning (PI-ML) (Karniadakis et al., 2021) opens new opportunities to combine the power of modern ML methods with physical constraints that serve as meaningful regularizers. These constraints might, for example, be available in the form of partial differential equations (PDEs). Within PI-ML we consider hydrodynamic flow problems governed by the physical law of mass conservation. This law is described in its local and differential form by a PDE commonly known as the *continuity equation* (CE)

$$\begin{cases} \partial_t \rho + \nabla \cdot (\boldsymbol{v}\rho) = 0 & (t, \boldsymbol{x}) \in (t_0, T) \times \Omega, \\ \rho(t_0, \boldsymbol{x}) = \rho_{t_0}(\boldsymbol{x}) & \boldsymbol{x} \in \Omega. \end{cases} \tag{1}$$

For any time $t \in [t_0, T)$ the function $\rho(t, \cdot)$ can be thought of as the density of parcels advected by the velocity field $\boldsymbol{v}$, with initial density $\rho_{t_0}$. Here, $[t_0, T] \times \Omega$ is the space-time domain, which is a subset of $\mathbb{R} \times \mathbb{R}^d$. The partial derivative w.r.t. time $t$ is denoted by $\partial_t$ and $\nabla \cdot \boldsymbol{b} = \nabla_{\boldsymbol{x}} \cdot \boldsymbol{b} = \sum_{i=1}^{d} \frac{\partial \boldsymbol{b}_i}{\partial x_i}$ is the spatial divergence of a $d$ dimensional vector field $\boldsymbol{b} : [t_0, T] \times \Omega \mapsto \mathbb{R}^d$.

Unlike classical initial value problems, we consider challenging settings where exact boundary and initial conditions are unknown. That is, the continuous density and velocity fields have to be inferred from sparse and noisy data. The important physical constraint is that the solution must comply with the CE in Eq. 1. To this end we propose a neural network based model that fulfills the CE by construction and provides physically consistent velocity and density fields.

We specifically consider two distinct application settings. In both settings we are restricted to sparse and noisy measurements. In setting (i) we measure the fluid density $\rho$ and velocity $\boldsymbol{v}$, without knowing any additional equations aside from the CE. This occurs for example within the area of radar ornithology (Chilson et al., 2017), where density and velocity of birds can be inferred from radar data. Such radar-based measurements are to date the only practical high-throughput data

source for birds. The goal is a spatiotemporal estimate of bird migration densities (Nussbaumer et al., 2019). Since more migration-specific dynamics are unknown, mass conservation is usually considered as a physical constraint (Nussbaumer et al., 2021; Lippert et al., 2022a;b). In setting (ii) we exclusively have sparse density observations, but we know additional equations constraining the velocity. This might, for example, occur in dynamical optimal transport problems. Here, two densities are to be interpolated while adhering to the CE. Therefore, minimizing the total transport cost constrains the velocity field. More generally, setting (ii) includes a wide range of compressible fluids dynamic problems. In both settings we are mainly interested in accurately modeling the density, with the velocity measurements or additional equations serving as an informative prior. We provide code at `https://github.com/bmda-unibas/LagrangianFlowNetworks`.

**Main contributions.** The main contributions of this paper are as follows:

- We outline a fundamental link between densities modeled by conditional Normalizing Flows and spatiotemporal density fields that satisfy the CE.
- We leverage this link to introduce models for ill-posed hydrodynamic flow problems that always satisfy the CE by construction, coined *Lagrangian Flow Networks* (LFlows). We do so without requiring an explicit representation of the initial density.
- We provide a way to calculate the velocity without inverting the conditional Normalizing Flow, enabling the use of flexible bijective layers with otherwise costly inverses.
- We assess LFlows in multiple application settings and show better predictive performance than existing methods while staying computationally feasible and physically consistent.

## 2 RELATED WORK

**Neural Networks for Conservation Laws.** Physics-informed neural networks (PINNs) (Raissi et al., 2019) enforce PDEs in neural networks by introducing an additional PDE-loss that penalizes pointwise deviations from the PDE. The PDE-loss is enforced on *collocation points* that are sampled in the signal domain. The accuracy of PINNs is thus limited by the amount (and distribution) of collocation points, as well as the dimension of the signal domain. For conservation laws, recent improvements in PINNs use more sophisticated sampling approaches (Arend Torres et al., 2022), or introduce domain decompositions (Jagtap et al., 2020). Although this alleviates scaling problems, the fundamental limitation due to the number of collocation points (or subdomains) still remains.

In contrast, Richter-Powell et al. (2022) propose a parameterization of neural networks that enforces mass conservation by design, which we refer to as *divergence-free Neural Networks* (DFNNs). Solutions to the CE in Eq. 1 are represented as divergence-free $(d + 1)$ dimensional vector field $\boldsymbol{b} = (\rho, \rho\boldsymbol{v})$ with an augmented $(d + 1)$ dimensional input space $\boldsymbol{s} = (t, \boldsymbol{x})$:

$$\frac{\partial \rho}{\partial t} + \nabla_{\boldsymbol{x}} \cdot (\rho\boldsymbol{v}) = \sum_{i=1}^{d+1} \frac{\partial b_i}{\partial s_i} = \nabla_{\boldsymbol{s}} \cdot \begin{pmatrix} \rho \\ \rho\boldsymbol{v} \end{pmatrix} = \nabla_{\boldsymbol{s}} \cdot \boldsymbol{b} = 0. \tag{2}$$

The generalization of divergence-free vector fields to higher dimensions is achieved through the concept of differential forms. The resulting parameterization, however, heavily relies on expensive higher-order automatic differentiation, posing limitations in terms of scalability.

Concurrent to our work, Li et al. (2023) (TIPF) proposes the use of Lagrangian flow maps to model continuous probability flows that fulfill the CE. Unlike LFlows, TIPF considers well-posed PDE settings with known initial conditions. Specifically, they focus on Fokker-Planck equations and Wasserstein gradient flows by additionally introducing an unbiased self-consistency loss. In contrast, we consider ill-posed data-assimilation for physical problems with sparse and noisy data where initial conditions are unknown. Additionally, LFlows stand out as they don't require the inversion of bijective layers for computing the velocity, allowing for more expressive bijections.

Deep operator learning (Lu et al., 2019; Li et al., 2020) was proposed as a general approach to learn dynamics from dense observations. In these settings, the PDE is not provided, but has to be learned from large amounts of dense data often provided through simulations. As such, it is not applicable to our setting with spatially sparse data obtained at irregular time steps. Further notable mentions are Lagrangian and Hamiltonian neural networks (Cranmer et al., 2020; Greydanus et al., 2019), which can learn conservation laws from observed trajectories of individual particles.

**Data assimilation with the Adjoint.** The adjoint method (Cacuci, 1981a;b; Pontryagin, 1987) allows to differentiate through numerical solvers by integrating adjoint equations backward in time. By minimizing an objective function, it is then possible to infer the initial conditions and parameters of a dynamical system from data. Within adjoint methods, the well-established semi-Lagrangian data assimilation (SLDA) approach is the closest one conceptually to our model and setting (Robert, 1982; Staniforth & Côté, 1991; Diamantakis & Magnusson, 2016). SLDA is widely used for for integrating transport equations into atmospheric models (Diamantakis, 2013; Hersbach et al., 2020). It is based on the Lagrangian form of the CE, which can be written as an ODE:

$$\left[ \begin{array}{c} \boldsymbol{x}(0, \boldsymbol{z}) \\ \ln \rho(0, \boldsymbol{z}) \end{array} \right] = \left[ \begin{array}{c} \boldsymbol{z} \\ \ln \rho_0(\boldsymbol{z}) \end{array} \right], \qquad \frac{\mathrm{d}}{\mathrm{d}t} \left[ \begin{array}{c} \boldsymbol{x}(t, \boldsymbol{z}) \\ \ln \rho(t, \boldsymbol{x}(t, \boldsymbol{z})) \end{array} \right] = \left[ \begin{array}{c} \boldsymbol{v}(t, \boldsymbol{x}(t, \boldsymbol{z})) \\ -\nabla \cdot \boldsymbol{v}(t, \boldsymbol{x}(t, \boldsymbol{z})) \end{array} \right], \quad (3)$$

where $\boldsymbol{z} \in \mathbb{R}^d$, $\boldsymbol{v} : [t_0, T) \times \mathbb{R}^d \mapsto \mathbb{R}^d, t \in (t_0, T)$. Given the initial position $\boldsymbol{z}$ and (log-) density $\ln \rho_0(z)$ of the parcel, the ODE describes their temporal evolution according to the CE. The density at the departure points (i.e. the initial time) is represented by an interpolated mesh. The data-loss for the density is computed by mapping observations backwards in time with Eq. 3 and then comparing it to the interpolated initial density.

An efficient autograd implementation of the adjoint method was presented by Chen et al. (2018). The implementation enables black-box differentiation for numerically solved ODEs, and further allows to specify the dynamics of ODEs with a neural network. The introduced Continuous Normalizing Flows can be regarded as a special case of the SLDA that propagates probability densities and models the velocity with a neural network, while fixing the density at the departure points. A limiting factor of neural adjoint based methods is their computational cost, since input derivatives of the velocity $\boldsymbol{v}$ are repeatedly evaluated for every step of the solver. Furthermore, dynamics given by neural networks can become stiff during training. To avoid these issues Biloš et al. (2021) propose time-dependent bijections instead of neural ODEs for modeling time series data.

**(Conditional) Normalizing Flows** Normalizing Flows (NFs) are a general approach to warping a simple probability distribution into a more complex target distribution via invertible and differentiable transformations, i.e. diffeomorphisms. Let $\boldsymbol{R} \in \mathbb{R}^d$ be a random variable with a known density function $\boldsymbol{R} \sim p_{\boldsymbol{R}}(\boldsymbol{r})$ and let $\boldsymbol{Y} = \mathcal{T}(\boldsymbol{R})$, where $\mathcal{T}$ is a diffeomorphism with trainable parameters. With a change of variables the probability density of $\boldsymbol{Y}$ can be expressed in terms of the *base density* $p_{\boldsymbol{R}}$, the map $\mathcal{T}$, and its Jacobian:

$$p_{\boldsymbol{Y}}(\boldsymbol{y}) = p_{\boldsymbol{R}}(\mathcal{T}^{-1}(\boldsymbol{y})) \big| \det J \mathcal{T}^{-1}(\boldsymbol{y}) \big|. \tag{4}$$

NFs usually rely on transformations for which the Jacobian determinant can be efficiently and easily calculated. A parameterization for conditional distributions $p_{\boldsymbol{Y}}(\boldsymbol{y}|\boldsymbol{c})$ can be obtained by additionally conditioning the parameters of $\mathcal{T}$ on another variable $\boldsymbol{c}$ through a hypernetwork (Ha et al., 2017). This is commonly called a conditional Normalizing Flow (Atanov et al., 2019; Kobyzev et al., 2020). For a review of NFs, we refer to Kobyzev et al. (2020) and Papamakarios et al. (2021).

## 3 LAGRANGIAN FLOW NETWORKS

We first present some key results of the classical theory of Lagrangian flows for smooth vector fields. These provide us a framework for evolving densities and velocities that always fulfill the CE. We then propose parameterizations that result in simple expressions for the velocity and density. The resulting LFlow models densities and velocities by building upon conditional Normalizing Flows.

### 3.1 FLOW MAPS AND THE CONTINUITY EQUATION

The Lagrangian view describes fluids from the perspective of moving fluid parcels, i.e. infinitesimal volumes with constant mass. From this point of view the CE states that density changes of the fluid are described by volume changes of parcels. That is, spatial contraction increases the density of a parcel, and expansion decreases it. In order to compute the density of any parcel, we then only need to know its initial density, and how much its volume was distorted.

More formally, let $\boldsymbol{x}_{t_0}$ denote the initial position of a parcel at time $t_0$. In addition, let $\boldsymbol{X}_t : \Omega \mapsto \Omega$ for a fixed $t \in [t_0, T]$ be a diffeomorphism that maps $\boldsymbol{x}_{t_0}$ to the parcel position at time $t$:

$$\boldsymbol{X}_t(\boldsymbol{x}_{t_0}) = \boldsymbol{x}_t, \tag{5}$$

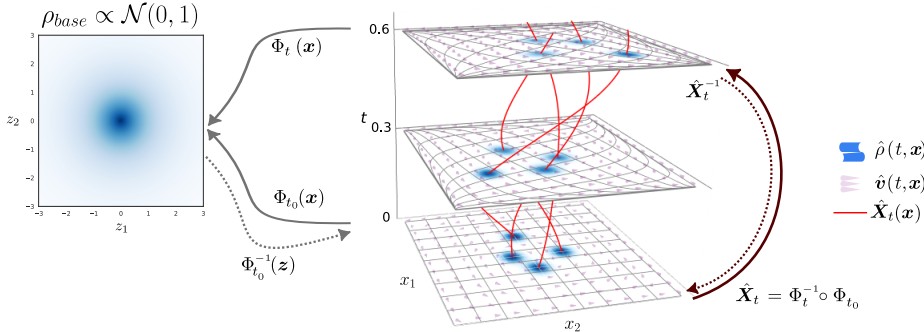

Figure 1: Illustration of the transformations and involved fields for modeling the temporal evolution of a 2D density with LFlows. The red lines indicate trajectories of fluid parcels.

with $\boldsymbol{X}_{t_0}$ being the identity map. That is, $\boldsymbol{X}_t$ provides the continuous trajectory of the parcel $\boldsymbol{x}_{t_0}$. We further assume basic regularity of $\boldsymbol{X}_t$ and $\boldsymbol{X}_t^{-1}$ such as smoothness and globally bounded derivatives. Since $\boldsymbol{X}_t$ provides the trajectory of a parcel, the velocity of a given parcel at position $\boldsymbol{x}_t$ and time $t$ follows naturally. First, the parcel is mapped back to its initial position with $\boldsymbol{X}_t^{-1}$. The velocity is then the change in position along its trajectory with respect to time $t$:

$$\boldsymbol{v}(t, \boldsymbol{x}) = \frac{\partial \boldsymbol{X}_t}{\partial t}\left(\boldsymbol{X}_t^{-1}(\boldsymbol{x})\right). \tag{6}$$

Following classical Cauchy Lipschitz theory, it is known that such a map $\boldsymbol{X}_t$ is the unique flow map of $\boldsymbol{v}$ starting at time $t_0$. Specifically, for any $\boldsymbol{x} \in \Omega$ the curve $t \mapsto \boldsymbol{X}_t(\boldsymbol{x})$ is the unique solution to the Cauchy Problem

$$\begin{cases} \partial_t \boldsymbol{X}_t(\boldsymbol{x}) = \boldsymbol{v}(t, \boldsymbol{X}_t(\boldsymbol{x})) & t \in [t_0, T), \\ \boldsymbol{X}_{t_0}(\boldsymbol{x}) = \boldsymbol{x}. \end{cases} \tag{7}$$

A more complete statement is given in A.1.1 Theorem 1, and we refer to Hartman (2002) for an extensive description of the theory of ordinary differential equations.

With the velocity given by Eq. 6, we further need to define a density to describe a fluid. Let $\rho_{t_0} : \Omega \mapsto \mathbb{R}_+$ be the (known) initial fluid density at time $t_0$. We can then define the time-evolved density as a transformation of $\rho_{t_0}$ using the change of variables formula:

$$\rho(t, \boldsymbol{x}) = \rho_{t_0}\left(\boldsymbol{X}_t^{-1}(\boldsymbol{x})\right)|\det J\boldsymbol{X}_t^{-1}(\boldsymbol{x})|. \tag{8}$$

Given the velocity in Eq. 6 and a smooth $\rho_{t_0}$, Eq. 8 is a solution to the continuity equation (see A.1.1 Theorem 2). Proofs for this statement vary significantly in their complexity and depend on the regularity assumptions for the velocity. That is, they range from classical theory to current mathematical research. For the sake of completeness, we provide precise statements, assumptions and proofs in the Appendix Section A.1. The appended proofs are based on the well-established classical theory for the existence and uniqueness of Lagrangian flows for smooth vector fields and we refer to Ambrosio & Crippa (2008) for basic and advanced results.

## 3.2 LAGRANGIAN FLOW NETWORKS

We can now exploit the derived connection between the CE and time-evolving diffeomorphisms to model densities and velocities that satisfy the CE by construction. Instead of directly parameterizing both $\boldsymbol{X}_t$ and $\rho_{t_0}$ (as in Li et al. (2023)), we model the density at each time $\rho_t$, including $\rho_{t_0}$, as a transformation of a simple fixed density $\rho_{\text{base}}$. This requires only one time-conditioned bijection $\Phi_t$. We call the resulting model Lagrangian Flow Networks (LFlows), which we illustrate in Figure 1.

Let $\Phi_t : \Omega \mapsto \mathbb{R}^d$ be a learnable diffeomorphism with $t \in [t_0, T]$. We propose to parameterize $\boldsymbol{X}_t$ as the composition

$$\hat{\boldsymbol{X}}_t(\boldsymbol{x}) = \Phi_t^{-1}\left(\Phi_{t_0}(\boldsymbol{x})\right). \tag{9}$$

In practice, we implement $\Phi_t$ as an invertible neural network with its parameters conditioned on time. We further define the initial density as a transformation of a simple base density:

$$\hat{\rho}_{t_0}(\boldsymbol{x}) = \rho_{\text{base}}\Big(\Phi_{t_0}(\boldsymbol{x})\Big) \left|\det J\Phi_{t_0}(\boldsymbol{x})\right|. \tag{10}$$

The base density $\rho_{\text{base}} : \mathbb{R}^d \mapsto \mathbb{R}_+$ is an unnormalized probability density:

$$\rho_{\text{base}}(\boldsymbol{z}) = c \cdot \mathcal{N}(\boldsymbol{0}, I), \tag{11}$$

where $c \in \mathbb{R}_+$ is the total mass of the system and a freely learnable parameter.

We now substitute $\boldsymbol{X}_t$ in Eq. 8 with the parameterized $\hat{\boldsymbol{X}}_t$ from Eq. 9. We further substitute the density $\rho_{t_0}$ with the parameterized $\hat{\rho}_{t_0}$ (Eq. 10). The modeled density then simplifies to

$$\hat{\rho}(t, \boldsymbol{x}) = \hat{\rho}_{t_0}\left(\hat{\boldsymbol{X}}_t^{-1}(\boldsymbol{x})\right) |\det J\hat{\boldsymbol{X}}_t^{-1}(\boldsymbol{x})| = \rho_{\text{base}}\Big(\Phi_t(\boldsymbol{x})\Big) \left|\det J\Phi_t(\boldsymbol{x})\right|. \tag{12}$$

We refer to the Appendix A.2.1 for the intermediate steps. Note that the resulting expression coincides with the change of variable formula for probability densities in Eq. 4. This allows us to elegantly model the evolving density through a conditional normalizing flow with unnormalized base density $\rho_{\text{base}}$ and bijective layers conditioned on time $\Phi_t$.

The parameterization of $\boldsymbol{X}_t$ with $\hat{\boldsymbol{X}}_t$ in Eq. 6 also results in a simple expression for the velocity:

$$\hat{\boldsymbol{v}}(t, \boldsymbol{x}) = \frac{\partial \hat{\boldsymbol{X}}_t}{\partial t}\left(\hat{\boldsymbol{X}}_t^{-1}(\boldsymbol{x})\right) = -\Big(J\Phi_t(\boldsymbol{x})\Big)^{-1} \frac{\partial \Phi_t}{\partial t}(\boldsymbol{x}). \tag{13}$$

We provide the explicit steps for arriving at Eq. 13 in the Appendix A.2.2. Note that in order to evaluate $\hat{\rho}$ and $\hat{\boldsymbol{v}}$ we now require only the forward map $\Phi_t$, but not its inverse. This proves useful for layers with expensive inverses, or if the inverse is unknown. An illustration that unifies the Lagrangian view and the provided parameterization of LFlows is given in Figure 1.

**Limitations.** LFlows model fluid densities and velocities by transforming a base density with bijective layers. As such, similar limitations as for Normalizing Flows apply. If the target density has disconnected modes, the base density must have the same number of disconnected modes due to topological constraints (Papamakarios et al., 2021). If not, the space in between disconnected modes will be covered by a small but non-zero density. Furthermore, LFlows are limited by the expressive power of the bijective layers. Even though state-of-the-art bijective layers are highly flexible, each layer might still be limited in terms of the number of modes that can be modeled (Liao & He, 2021).

## 4 IMPLEMENTATION

To implement LFlow as outlined in Section 3, we require conditional bijective layers. That is, the diffeomorphism $\Phi_t$ required for Eq. 12 and Eq. 13 has to be conditioned on time $t$. To allow for a flexible parameterization, we first embed $t$ into a higher-dimensional space with an embedding network and then condition the bijections on this embedding, i.e. $\Phi_t := \Phi(\boldsymbol{x}; f_\Theta(t))$ with $f_\Theta : [t_0, T] \mapsto \mathbb{R}^k$. The $k$-dimensional embedding is shared between the individual layers. A high-level visualization of the network architecture is provided in the Appendix Figure A.6. We implement $f_\Theta(t)$ as MLPs with residual skip connections and swish activations (Elfwing et al., 2018).

We implement flexible $\Phi(\boldsymbol{x}; f_\Theta(t))$ with Lipschitz-constrained invertible densenets (Perugachi-Diaz et al., 2021), which have free-form Jacobians. For a conditional variant of these layers we pass the embedding $f_\Theta(t)$ as an additional input. That is, each invertible densenet layer is a function $g(\boldsymbol{x}, t) = \boldsymbol{x} + h(\boldsymbol{x}, f_\Theta(t))$ with $h : \mathbb{R}^{d+k} \mapsto \mathbb{R}^d$ and $Lip(h) < 1$. The activations of $h$ are sinusoidal $s(x) = \sin(\omega x)/\omega$ with $\omega \in \mathbb{R}_+$, where the division by $\omega$ ensures $Lip(s) = 1$. Invertible densenets provide numerical inversion via fixed-point iterations. In addition to densenet blocks, we employ (unconditional) intermediate activation normalizations (Kingma & Dhariwal, 2018) and (conditional) SVD layers. The orthogonal components of the SVD layers are parameterized by conditional Householder reflections.

**Normalization Constant.** In all our settings the total mass, i.e. the normalization constant $c$ in Eq. 11, is not known. Therefore, we treat $c$ as a freely learnable hyperparameter, which we initialize based on a validation set. We further encourage solutions with a small total mass $c$ during training with the penalty $L_{\text{mass}} = w_c \cdot c$, where $w_c \in \mathbb{R}_{\geq 0}$ is a hyperparameter. In practice, this penalty discourages learning significant densities in areas where there are no measurements.

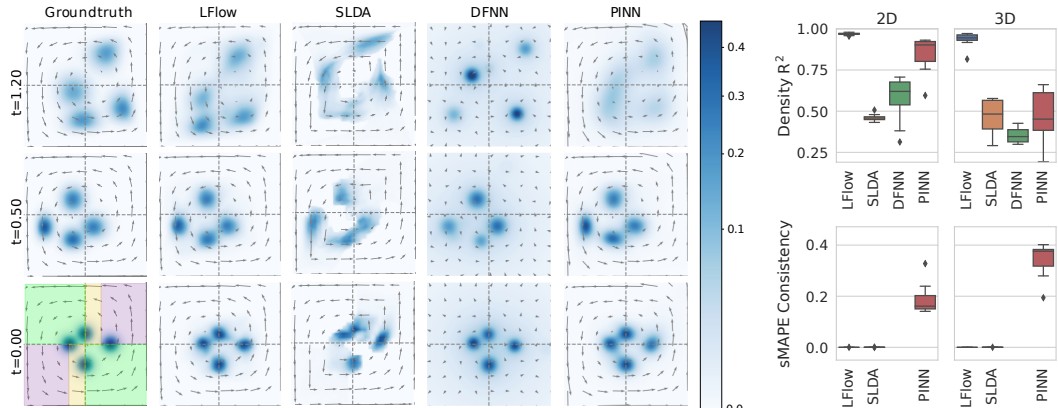

Figure 2: Evaluation of the predicted density on the $xy$-plane with $z = 0$ for different methods. Arrows indicate velocity, except for the DFNNs, where the normalized flux is shown. The lower left plot shows the spatial splitting into train (green), validation (yellow), and test (purple) subsets.

**Baseline Models.** For implementation details of all competing methods, we refer to the Appendix Section A.3 and provided code.

## 5 EXPERIMENTS

We showcase LFlows for two distinct settings. In setting (i), density and velocity are observed, but no equations are available (Section 5.1 and 5.3). In setting (ii) only the density is observed but further equations for the velocity are known (Section 5.2). For details on the data, architecture, and code for each experiment, we refer to the Appendix A.4 to A.6 and the supplementary material. We will compare LFlows with methods that enforce the CE through different means, namely divergence-free neural networks (DFNNs), physics-informed neural networks (PINNs), and semi-Lagrangian data assimilation (SLDA). For details on the implementation we refer to the Appendix A.3.

**Numerical Evaluation of Physical Consistency.** Physical consistency in terms of the CE implies that the predicted density at any time coincides with the initial density transformed forward in time by the learned velocity field. An inconsistent model would imply that the predicted velocity field fundamentally disagrees with the predicted density movements. This would make any downstream interpretation of the two learned fields futile. We quantitatively evaluate this potential inconsistency in the following experiments. We compare the predicted density $\hat{\rho}_{\text{model}}(t, \boldsymbol{x})$ with the numerical solution of the IVP defined by $\hat{\rho}_{\text{model}}(t_0, \boldsymbol{x})$, $\hat{\boldsymbol{v}}_{\text{model}}(t, \boldsymbol{x})$, and Eq. 3. We evaluate the symmetric mean absolute percentage error $\text{sMAPE} = \frac{1}{n} \sum_{i=1}^{n} \frac{|\hat{\rho}_{\text{model}}(t_i, \boldsymbol{x}_i) - \hat{\rho}_{\text{ODE}}(t_i, \boldsymbol{x}_i)|}{|\hat{\rho}_{\text{model}}(t_i, \boldsymbol{x}_i)| + |\hat{\rho}_{\text{ODE}}(t_i, \boldsymbol{x}_i)|}$ that ranges from 0 to 1.

### 5.1 SIMULATED FLUID FLOW

For a synthetic example of setting (i) we simulate densities in 2D and 3D over time by transforming a mixture of four unnormalized Gaussians. We parameterize time-dependent bijections in $t \in [0, 1.2], \Omega = (-4, 4)^d$, which provide us analytical forms for the densities and velocities. During training, only sub-regions of the domain $\Omega$ are observed. The dynamics in 3D are similar to the 2D setting, with the $xy$-velocity being the same for all $z$ values and the $z$ velocity being 0. The only added difficulty is a higher-dimensional domain. We limit all models to the computing resources of a NVIDIA Titan X Pascal, and optimize based on the explained variance[1] ($R^2$) of the density on the validation set. For the PINN this resulted in $2^{16}$ collocation points. We do not include consistency results for DFNNs due to numerical instabilities. As DFNNs only provide access to the flux $\hat{\boldsymbol{F}}$, the velocity $\hat{\boldsymbol{v}} = \hat{\boldsymbol{F}}/\hat{\rho}$ becomes numerically unstable in low-density regions, which are abundant in this experiment.

---

[1]$R^2 = 1 - \frac{MSE(y_{obs}, \hat{y})}{Var(y_{obs})} \leq 1$ with $R^2 = 1$ indicating a perfect reconstruction.

Results for 10 random seeds are provided in Figure 2. In addition, snapshots of the 3D prediction for $z = 0$ are shown. All methods aside from the PINN have a low consistency sMAPE on the order of 1e-4 or lower. This is expected, as LFlows enforce the CE by construction. Furthermore, SLDA computes the density similarly to our numerical reference, although with a lower order ODE solver. Low error tolerances or low-order solvers for SLDA would of course still result in inconsistencies. Looking at the predictive performance, LFlows show the highest average $R^2$ for the density in both 2D and 3D. While PINNs perform competitively in 2D, they severely degrade in 3D, as the number of collocation points used (limited by GPU memory) is insufficient to enforce the PDE. This is also reflected in their increase of the consistency sMAPE in 3D. Finally, DFNNs and SLDA are roughly comparable in terms of predictive accuracy, with DFNNs being in our experience most prone to overfitting. We note that small density displacements can already lead to large differences in $R^2$.

## 5.2 DYNAMICAL OPTIMAL TRANSPORT

As an example of setting (ii), in which no velocity is observed but additional equations dictating the dynamics are known, we consider dynamical optimal transport problems. Our experimental setting closely follows Richter-Powell et al. (2022). Specifically, we consider the Benamou-Brenier formulation of the optimal transport problem between two densities $p_{t_0}$ and $p_{t_1}$. In this case the optimal transport map is the solution map $\boldsymbol{X}_t$ of a flow that is defined by the vector field $\boldsymbol{v}$ and minimizes the following objective:

$$\min_{\boldsymbol{v}, \rho} \int_{t_0}^{t_1} \int_{\Omega} |\boldsymbol{v}(t, \boldsymbol{x})|^2 \rho(t, \boldsymbol{x}) \, dx \, dt \tag{14}$$

subject to the constraints $\rho(t_0, \boldsymbol{x}) = p_{t_0}(\boldsymbol{x})$ and $\rho(t_1, \boldsymbol{x}) = p_{t_1}(\boldsymbol{x})$. Furthermore, $\rho$ and $\boldsymbol{v}$ are subject to the continuity equation $\partial_t \rho = -\nabla \cdot (\rho \boldsymbol{v})$.

Both DFNNs and LFlows can solve the minimization problem in Eq. 14 without needing a separate estimation of $\rho$. Instead, one fits the densities at $t_0$ and $t_1$ and additionally minimize Eq. 14. However, to obtain the transport map from the learned velocity field, DFNNs need to numerically solve the Cauchy problem in Eq. 7. An ODE solver might however struggle due to the numerical instability of the DFNNs velocity in low-density regions. In contrast, LFlows elegantly provide an analytical form for the continuous transport map through the learned bijections, i.e. $\hat{\boldsymbol{X}}_t(\boldsymbol{x}) = \Phi_t^{-1}(\Phi_{t_0}(\boldsymbol{x}))$.

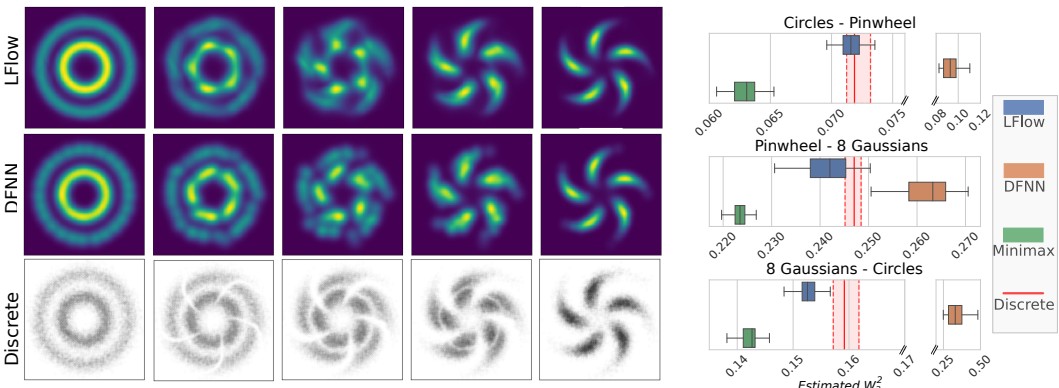

Figure 3: *Left:* Approximations of the 2D optimal transport map with LFlows, DFNN and a discrete reference. *Right:* Corresponding estimated Wasserstein distances for different methods for a single run. The red vertical lines denote the minimum, median, and maximum estimates of 5 runs with a discrete OT solver. Datasets: Circles ↔ Pinwheel, Pinwheel ↔ 8Gaussians, 8Gaussians ↔ Circles.

We train the models by optimizing

$$\min_{\hat{\rho}, \hat{\boldsymbol{v}}} \lambda \mathbb{E}_{\tilde{p}_0} \left[ |\hat{\rho}(0, \boldsymbol{x}) - p_0(\boldsymbol{x})| \right] + \lambda \mathbb{E}_{\tilde{p}_1} \left[ |\hat{\rho}(1, \boldsymbol{x}) - p_1(\boldsymbol{x})| \right] + \int_0^1 \int_{\Omega} |\hat{\boldsymbol{v}}(t, \boldsymbol{x})|^2 \hat{\rho}(t, \boldsymbol{x}) \, dx \, dt \tag{15}$$

where data is drawn from $\tilde{p}_i$, which is a mixture of $p_i$ and a uniform density (for $i = 0, 1$); $\lambda$ is a hyperparameter. At test time we empirically estimate the $W_2^2$ distance by mapping 5000 samples of $p_0$ from $t = 0$ to $t = 1$. Different to Richter-Powell et al. (2022) we repeat this estimate 50 times. We compare the $W_2^2$ estimates of (i) LFlows, (ii) DFNNs and (iii) a minimax formulation of the optimal transport map learned via input convex neural network (Makkuva et al., 2020). For DFNNs the samples are transported through the estimated velocity with an ODE solver. We further compare to the $W_2^2$ estimates of a discrete OT-solver (Bonneel et al., 2011) from the *pot* library (Flamary et al., 2021) based on 50000 samples. The $\lambda$ for LFlows is chosen by matching the $W_2^2$ distance between the *moons* and *swissroll* datasets with the discrete estimate. For DFNNs a $\lambda$ value is provided by Richter-Powell et al. (2022). In this experiment we restrict ourselves to methods that can exactly enforce the CE and thus exclude PINNs.

We considered three different pairs of toy 2D distributions. Figure 3 shows the approximated optimal transport maps learned with LFlows and the $W_2^2$ estimates of the different methods. We verified that the DFNN and the LFlow fit the target densities well, with a test MSEs below 8e-5 for all settings. We excluded SLDA, as it was unstable and did not consistently result in low errors for the two target densities. The LFlow estimates of $W_2^2$ are closest to the range of discrete estimates (Figure 3). In contrast, the minimax model underestimates the distance which is consistent with the results of Richter-Powell et al. (2022). DFNNs significantly overestimated $W_2^2$ and we were unable to fully reproduce results obtained by Richter-Powell et al. (2022). We assume that this is due to the ODE solver struggling with the unstable velocity calculation in low-density regions.

### 5.3 MODELING BIRD MIGRATION

As a real-world application for setting (i) we model bird migration within Europe based on weather radar measurements. The data provided by Nussbaumer et al. (2021) contains estimated bird densities ($birds/km^3$) and velocities ($m/s$). Measurements are taken from 37 weather radar stations in France, Germany, and the Netherlands at up to 5-minute intervals at $200m$ altitude bins, reaching up to $5km$. The velocity data does not include a $z$-axis component. We model the bird migration of 3 subsequent nights of April 2018. We assume that the mass is mostly conserved within the three nights during migration. We test the predictions on radars located in the center of the covered region, which were excluded during training (see Figure 4). Hyperparameters of all models are selected by minimizing the density MSE on three nights of March 2018. As a baseline, we compare the results with a 10-layer multilayer perceptron (MLP) with skip connections, 256 hidden units per layer, ReLU activations, and Batch normalization. We additionally evaluate the PINN, DFNN, and SLDA. The PINN has the same general architecture as the MLP but additionally minimizes the penalty $\lambda \cdot \|\partial_t \hat{\rho} + \nabla \cdot (\hat{\rho}\hat{v})\|_2$ evaluated on 100 000 collocation points, where $\lambda \in \mathbb{R}_+$ is a hyperparameter.

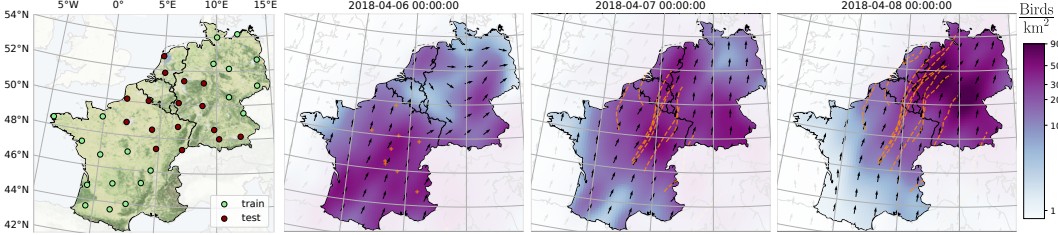

Figure 4: Snapshots of predicted bird density at three consecutive nights within central Europe. The 2D projection was obtained by integrating over altitudes covered by the radars. Orange lines indicate 2D ($xy$) projections of 3D trajectories from $t_0$ to $t$ using randomly sampled departure points.

Figure 4 shows snapshots of the vertically integrated density and flux predicted by LFlows. Predictions of the other models are provided in the Appendix A.6.2. Aside from the velocity and density, LFlows readily provide the trajectories (shown in orange). In practice, experts could compare these with the migration paths taken by individual birds, which are for example obtained from bird-ringing studies. We further explore the role of the total mass penalty $w_c$ in Appendix A.6.3.

We compare test density errors for each model in Figure 5 (c). Methods enforcing the CE result in a lower error than the baseline MLP. The consistency sMAPE shows the PINN and MLP lead to

inconsistent density and velocity. LFlows, SLDA and DFNNs again have a sMAPE that is on the order of 1e-4 or lower. Figure 5 (a) and (b) show the pointwise sMAPE for the PINN and LFlow.

While DFNNs and SLDA are nearly competitive with LFlows in terms of the density MSE, they suffer from high computational costs. SLDA requires to evaluate a neural network many times to numerically solve the ODE. In addition, SLDA gets slower during training because the dynamics given by the neural network become more stiff. This is shown by the increasing runtime for each epoch in Figure 5 (e) and is a known limitation of models of the neural ODE family (Biloš et al., 2021). DFNNs on the other hand result in huge memory requirements due to the required second order derivatives. Figure 5 (f) shows the peak memory use in terms of VRAM for varying minibatch sizes of the models. In practice, high memory requirements result in small minibatch sizes, which ultimately lead to a slower training and inference pipeline (Shallue et al., 2018). The high peak memory and runtime of PINNs is due to the large amount of collocation points.

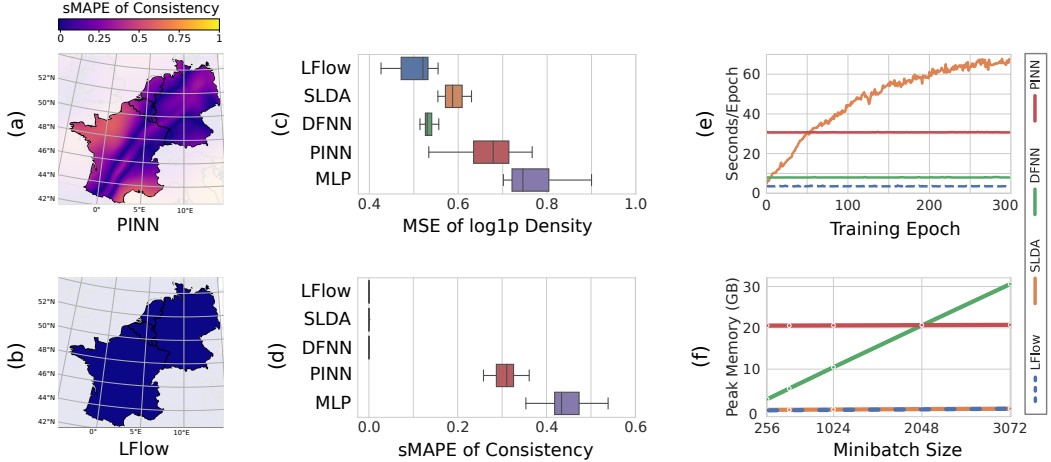

Figure 5: Results of the bird migration experiment. *(a)* Consistency sMAPE of PINNs and *(b)* LFlows at a fixed time (2018-04-08 00:00). *(c)* Test MSE of the (log1p) density and *(d)* consistency sMAPE evaluated at multiple timesteps. *(e)* Time per training epoch during training and *(f)* peak memory usage during training in GB VRAM for varying minibatch sizes.

## 6   CONCLUSION

We introduced LFlows for modeling densities and velocities that adhere to the continuity equation by construction. We did so by establishing a link between time-conditioned diffeomorphisms and Lagrangian solution maps for the continuity equation. The resulting parameterization allows us to elegantly model time evolving densities with a single time-conditioned Normalizing Flow. Furthermore, we can calculate the velocity without inverting the conditional bijections, allowing the use of expressive bijective layers. We showed that LFlows can be applied to settings where we have sparse data on both density and velocity, and to settings where we have no data on velocity, but instead enforce additional equations.

In terms of density prediction LFlows outperform all competing models on both synthetic and real experiments. Different to methods like PINNs, which weakly enforce the continuity equation, LFlows always provide physically consistent predictions. In addition, LFlows avoid scaling limitations of DFNNs (peak memory usage) and neural adjoint based methods (training time). For downstream tasks, LFlows directly provide Lagrangian maps without the need for additional numerical solvers. In dynamical optimal transport settings, LFlows directly provide the analytic expression of the transport map. When modeling bird migration, the Lagrangian maps provide access to trajectories, which could be compared to migration paths obtained from different data modalities.

## 7 Reproducibility Statement

We provide general information on our implementation of LFlows, DFNNs, SLDA and PINNs in Section 4 and in the Appendix A.3. For each experiment we provide further details in the Appendix A.4 to A.6, where we state the layers, settings, and computational ressources (GPU) used. We also describe how we generated the introduced synthetic dataset. For the real-world bird-migration dataset we provided references for accessing the data, as well as information on the preprocessing in the Appendix A.6. The supplementary material contains code for all experiments and models. The `readme.md` file lists the random seeds used for experiments and plots. The selected hyperparameters for each experiment are provided. The code further includes the generation of the synthetic datasets, as well as automated scripts for downloading and preprocessing the bird-migration data. A cleaned anaconda environment file for reproducing the python environment is provided.

We furthermore provide access to the used conditional bijective layers in a separate python package called *Flow Conductor*[2]. This package includes conditional i-DenseNets with sinusoidal activations, as well as the conditional SVD layers.

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

# A    APPENDIX

## A.1    THEORETICAL BACKGROUND

In this section we provide proofs and theoretical background for the method in Section 3. While the underlying theory is well established, we were unable to find a single source that concisely contained all required statements written in an accessible and directly citeable manner. Section A.1.1 contains the statements we rely on in Section 3. The proofs for these theorems are provided in Sections A.1.2 to A.1.4

### A.1.1    TIME DEPENDENT BIJECTIONS AND THE CONTINUITY EQUATION

Theorem 1 provides us a velocity field given time-dependent bijections. Theorem 2 links the push-forward of the density to the solution of the continuity equation defined by an initial density and the velocity given by Theorem 1.

**Theorem 1.** *Let $0 \leq t_0 < T$ and let $\Omega \subset \mathbb{R}^d$ be a convex open set. Let $\boldsymbol{X} : [t_0, T] \times \Omega \to \Omega$ be a family of maps such that $\boldsymbol{X}_t : \Omega \to \Omega$ is a bijection for any $t \in [t_0, T]$ and $\boldsymbol{X}_{t_0}(\boldsymbol{x}) = \boldsymbol{x}$ for any $\boldsymbol{x} \in \Omega$. Assume that $\boldsymbol{X}$ and $\boldsymbol{X}^{-1}$ are $C^\infty([t_0, T] \times \Omega; \Omega)$ with globally bounded derivatives.*

*Then, the velocity field $\boldsymbol{v}(t, \boldsymbol{x}) = \frac{\partial \boldsymbol{X}_t}{\partial t}\big(\boldsymbol{X}_t^{-1}(\boldsymbol{x})\big)$ is $C^\infty$. In particular, $\boldsymbol{v}$ satisfies the assumptions of the Cauchy–Lipschitz Theorem 3 and $\boldsymbol{X}$ is the unique flow map of $\boldsymbol{v}$ starting at time $t_0$. Specifically, for any $\boldsymbol{x} \in \Omega$ the curve $t \mapsto \boldsymbol{X}_t(\boldsymbol{x})$ is the unique solution to the Cauchy Problem*

$$\begin{cases} \partial_t \boldsymbol{X}_t\,(\boldsymbol{x}) = \boldsymbol{v}\,(t, \boldsymbol{X}_t(\boldsymbol{x})) & t \in [t_0, T), \\ \boldsymbol{X}_{t_0}(\boldsymbol{x}) = \boldsymbol{x} \end{cases} \tag{16}$$

*Proof.* See Appendix Section A.1.3.

**Theorem 2.** *Let $\Omega, T, t_0, \boldsymbol{X}$ be as in Theorem 1. Given an initial density $\rho_{t_0} \in L^1(\Omega)$, we define*

$$\rho(t, \boldsymbol{x}) = \rho_{t_0}(\boldsymbol{X}_t^{-1}(\boldsymbol{x}))|\det J\boldsymbol{X}_t^{-1}(\boldsymbol{x})|. \tag{17}$$

*Then $\rho(t, \boldsymbol{x})$ is a distributional solution to the continuity equation in Eq. 1 according to Definition 1, i.e. the following condition is satisfied for any test function $\phi \in C_c^\infty([t_0, T) \times \Omega)$:*

$$\int_{t_0}^T \int_\Omega (\partial_t \phi + \boldsymbol{v} \cdot \nabla\phi)\rho \, dx \, dt = - \int_\Omega \rho_{t_0}(\boldsymbol{x})\phi(t_0, \boldsymbol{x}) \, dx. \tag{18}$$

*Moreover, if $\rho_{t_0} \in C^\infty(\Omega)$, then $\rho \in C^\infty([t_0, T) \times \Omega)$ and $\rho$ is a point-wise solution to the continuity equation Eq. 1. If we assume in addition that $\rho_{t_0}(\boldsymbol{x}) > 0$ for any $\boldsymbol{x} \in \Omega$, then the same holds for $\rho(t, \boldsymbol{x})$ for any $(t, \boldsymbol{x}) \in [t_0, T) \times \Omega$ and $\rho$ satisfies the log-density formula of the continuity equation*

$$\frac{d}{dt} \log(\rho(t, \boldsymbol{X}_t(\boldsymbol{x}))) = -\nabla \cdot \boldsymbol{v}(t, \boldsymbol{X}_t(\boldsymbol{x})). \tag{19}$$

*Proof.* See Appendix Section A.1.2 and A.1.4.

### A.1.2    THE FLOW ASSOCIATED TO A LIPSCHITZ VECTOR FIELD

We recall the setting of the classical Cauchy–Lipschitz Theorem. For simplicity, let $\Omega \subset \mathbb{R}^d$ be a convex open set. Given $0 \leq t_0 < T$, let $\boldsymbol{v} : [t_0, T] \times \Omega \to \mathbb{R}^d$ be a bounded vector field. We say that $\boldsymbol{v}$ is Lipschitz continuous in space uniformly in time if there exists a constant $L > 0$ such that

$$|\boldsymbol{v}(t, \boldsymbol{x}) - \boldsymbol{v}(t, \boldsymbol{y})| \leq L|\boldsymbol{x} - \boldsymbol{y}| \quad \forall t \in [t_0, T] \, \forall \boldsymbol{x}, \boldsymbol{y} \in \Omega. \tag{20}$$

Throughout this section, we consider maps $\boldsymbol{X} : [t_0, T] \times \Omega \to \Omega$ with the following properties:

- for any $\boldsymbol{x} \in \Omega$ the map $t \mapsto \boldsymbol{X}_t(\boldsymbol{x})$ is $C^1([t_0, T])$ with uniform bounds, namely there exists a constant $M > 0$ such that

$$|\partial_t \boldsymbol{X}_t(\boldsymbol{x})| \leq M \quad \forall t \in [t_0, T] \, \forall \boldsymbol{x} \in \Omega; \tag{21}$$

- for any $t \in [t_0, T]$ the map $\boldsymbol{x} \mapsto \partial_t \boldsymbol{X}_t(\boldsymbol{x})$ is Lipschitz with uniform bounds, namely there exists a constant $L > 0$ such that

$$|\partial_t \boldsymbol{X}_t(\boldsymbol{x}) - \partial_t \boldsymbol{X}_t(\boldsymbol{y})| \leq L|\boldsymbol{x} - \boldsymbol{y}| \quad \forall t \in [t_0, T] \ \forall \boldsymbol{x}, \boldsymbol{y} \in \Omega; \tag{22}$$

- for any $t \in [t_0, T]$ the map $\boldsymbol{x} \mapsto \boldsymbol{X}_t(\boldsymbol{x})$ is a bilipschitz transformation of $\Omega$ uniformly in time, namely $\boldsymbol{X}_t$ is a bijection of $\Omega$ and there exists a constant $C > 0$ such that

$$C^{-1}|\boldsymbol{x} - \boldsymbol{y}| \leq |\boldsymbol{X}_t(\boldsymbol{x}) - \boldsymbol{X}_t(\boldsymbol{y})| \leq C|\boldsymbol{x} - \boldsymbol{y}| \quad \forall \boldsymbol{x}, \boldsymbol{y} \in \Omega \ \forall t \in [t_0, T]. \tag{23}$$

We state the Cauchy–Lipschitz Theorem in the case of $\mathbb{R}^d$ and for the forward flow. We refer to Hartman (2002) for an extensive description of the theory of ordinary differential equations, as well as any book in basic differential calculus.

**Theorem 3.** *Given $T > 0$ and $t_0 \in [0, T)$, let $\boldsymbol{v}: [t_0, T] \times \mathbb{R}^d \to \mathbb{R}^d$ be a bounded vector field that satisfies Eq. 20 for some constant $L > 0$. For any $\boldsymbol{x} \in \mathbb{R}^d$ there exists a unique trajectory $t \mapsto \boldsymbol{X}_t(\boldsymbol{x})$ solving the Cauchy problem*

$$\begin{cases} \partial_t \boldsymbol{X}_t(\boldsymbol{x}) = \boldsymbol{v}(\boldsymbol{X}_t(\boldsymbol{x}), t) & t \in [t_0, T), \\ \boldsymbol{X}_{t_0}(\boldsymbol{x}) = \boldsymbol{x} \end{cases} \tag{24}$$

*in the integral sense. The map $\boldsymbol{X} : [t_0, T) \times \mathbb{R}^d \to \mathbb{R}^d$ is the flow of $\boldsymbol{v}$ starting at time $t_0$ and it satisfies Eq. 21, Eq. 22, Eq. 23.*

**Remark 1.** *Under the assumptions of Theorem 3, we point out that the maps $\boldsymbol{X}_t, \boldsymbol{X}_t^{-1}$ are Lipschitz continuous for any time. Thus, given a time slice $t \in [t_0, T)$, we infer that $\boldsymbol{X}_t, \boldsymbol{X}_t^{-1}$ are differentiable almost everywhere in $\mathbb{R}^d$. Hence, the Jacobian matrices $J\boldsymbol{X}_t(\boldsymbol{x}), J\boldsymbol{X}_t^{-1}(\boldsymbol{x})$ are well defined for almost every $\boldsymbol{x} \in \mathbb{R}^d$ and we have that*

$$J\boldsymbol{X}_t(\boldsymbol{X}_t^{-1}(\boldsymbol{x})) = [J\boldsymbol{X}_t^{-1}(\boldsymbol{x})]^{-1} \quad \text{for almost every } \boldsymbol{x} \in \mathbb{R}^d.$$

*Moreover, there exists a constant $M > 0$ such that*

$$|J\boldsymbol{X}_t(\boldsymbol{x})| + |J\boldsymbol{X}_t^{-1}(\boldsymbol{x})| \leq M \quad \text{for almost every } \boldsymbol{x} \in \mathbb{R}^d.$$

*Here, $|\cdot|$ is a given matrix norm (recall that all norms are equivalent in finite dimensional vector spaces). We point out that the uniqueness part of Theorem 3 and the regularity properties of the flow, as well as the existence of the Jacobian matrix, extend to Lipschitz vector field defined on a general convex domain $\Omega$, as soon as we assume that the trajectories do not touch the boundary of $\Omega$. In this case, we get that the flow map is a bilipschitz transformation of $\Omega$ for any time slice.*

### A.1.3 VELOCITIES FROM 1-PARAMETER GROUPS OF DIFFEOMORPHISM

Theorem 1 is a particular case of the following more general result. For simplicity, we consider convex domains.

**Theorem 4.** *Let $0 \leq t_0 < T$, let $\Omega \subset \mathbb{R}^d$ be a convex open set and let $\boldsymbol{X} : [t_0, T] \times \Omega \to \Omega$ be a family of maps on $\Omega$ such that $\boldsymbol{X}_{t_0}(\boldsymbol{x}) = \boldsymbol{x}$ for any $\boldsymbol{x} \in \Omega$. Assume that $\boldsymbol{X}$ satisfies Eq. 21, Eq. 22 and Eq. 23. Then, the velocity field $\boldsymbol{v}(t, \boldsymbol{x}) = \frac{\partial \boldsymbol{X}_t}{\partial t}(\boldsymbol{X}_t^{-1}(\boldsymbol{x}))$ satisfies the assumptions of the Cauchy–Lipschitz Theorem 3 and $\boldsymbol{X}$ is the unique flow map of $\boldsymbol{v}$ starting at time $t_0$. Specifically, for any $\boldsymbol{x} \in \Omega$ the curve $t \mapsto \boldsymbol{X}_t(\boldsymbol{x})$ is the unique solution to the Cauchy problem Eq. 16.*

The reader might note that Theorem 4 is the inverse of Theorem 3. Indeed, given a map $\boldsymbol{X}$ satisfying all the properties of a flow map, it is natural to ask whether we can find a velocity field $\boldsymbol{v}$ within the Cauchy–Lipschitz framework whose flow starting at time $t_0$ is $\boldsymbol{X}$.

*Proof of Theorem 1.* Given $(t, \boldsymbol{x}) \in [t_0, T) \times \Omega$, we define

$$\boldsymbol{v}(t, \boldsymbol{x}) = \lim_{h \to 0} \frac{\boldsymbol{X}_{t+h}(\boldsymbol{X}_t^{-1}(\boldsymbol{x})) - \boldsymbol{X}_t(\boldsymbol{X}_t^{-1}(\boldsymbol{x}))}{h} = \partial_t \boldsymbol{X}_t(\boldsymbol{X}_t^{-1}(\boldsymbol{x})). \tag{25}$$

Since the map $s \mapsto \boldsymbol{X}_s(\boldsymbol{X}_t^{-1}(\boldsymbol{x}))$ is $C^1([t_0, T])$ by assumption for any $\boldsymbol{x} \in \Omega$, the velocity field $\boldsymbol{v}$ is well defined. Moreover, by Eq. 25, the curve $t \mapsto \boldsymbol{X}_t(\boldsymbol{x})$ solves the Cauchy problem Eq. 24 with $\boldsymbol{v}$ given by Eq. 25. We study the regularity of $\boldsymbol{v}$ to ensure that $\boldsymbol{X}$ is the *unique* flow associated to $\boldsymbol{v}$ starting at time $t_0$. To begin, we remark that $\boldsymbol{v}$ is uniformly bounded by Eq. 21. Moreover, since $\boldsymbol{X}_t^{-1}, \partial_t \boldsymbol{X}_t$ are Lipschitz maps for any time slice $t \in [t_0, T]$ by Eq. 22 and Eq. 23, we infer that $\boldsymbol{v}$ satisfies Eq. 20. Thus, $\boldsymbol{v}$ satisfies the assumptions of Theorem 3. In particular, $\boldsymbol{X}$ is the unique flow starting at time $t_0$ of the vector field $\boldsymbol{v}$. $\qquad\square$

A.1.4  THE CONTINUITY EQUATION

Let $\Omega \subset \mathbb{R}^d$ be an open set, let $T > 0$ and $t_0 \in [0, T)$. Given a vector field $\boldsymbol{v} : [t_0, T] \times \Omega \to \mathbb{R}^d$, we shall consider the continuity equation Eq. 1 on $(t_0, T) \times \Omega$ with initial condition $\rho_{t_0}$. To deal with irregular vector fields and densities, we consider solutions to Eq. 1 in the sense of distributions. We refer to Ambrosio & Crippa (2008) for some basic and advanced results on the theory of continuity equation.

**Definition 1.** *Let $\Omega \subset \mathbb{R}^d$ be an open set and $0 \leq t_0 < T$. Let $\boldsymbol{v} \in L^\infty([t_0, T] \times \Omega; \mathbb{R}^d)$ be a vector field and let $\rho_{t_0} \in L^1(\Omega)$. We say that $\rho \in L^\infty([t_0, T]; L^1(\Omega))$ is a distributional solution to Eq. 1 if the following condition is satisfied for any test function $\phi \in C_c^\infty([t_0, T) \times \Omega)$:*

$$\int_{t_0}^T \int_\Omega (\partial_t \phi + \boldsymbol{v} \cdot \nabla \phi) \rho \, dx \, dt = - \int_\Omega \rho_{t_0}(x) \phi(t_0, x) \, dx. \tag{26}$$

**Remark 2.** *We point out that Definition 1 is well posed without differentiability assumptions on $\boldsymbol{v}, \rho$. However, if $\boldsymbol{v}, \rho$ are $C^1$ functions in time and space and $\rho_{t_0}$ is a continuous function, after integrating by parts the left hand side of Eq. 26, for any test function $\phi \in C_c^\infty([t_0, T) \times \Omega)$ we obtain that*

$$\int_{t_0}^T \int_\Omega (\partial_t \rho + \nabla \cdot (\boldsymbol{v}\rho)) \phi \, dx \, dt + \int_\Omega \rho(t_0, \boldsymbol{x}) \phi(t_0, x) \, dx = \int_\Omega \rho_{t_0}(\boldsymbol{x}) \phi(t_0, \boldsymbol{x}) \, dx.$$

*Since $\partial_t \rho + \nabla \cdot (\boldsymbol{v}\rho)$ is a continuous function, by the so-called Fundamental Lemma of Calculus of Variations (see Brezis (2011), for instance) we infer that $\partial_t \rho + \nabla \cdot (\boldsymbol{v}\rho) = 0$ for any $(t, \boldsymbol{x}) \in (t_0, T) \times \Omega$ and $\rho_{t_0}(\boldsymbol{x}) = \rho(t_0, \boldsymbol{x})$ for any $\boldsymbol{x} \in \Omega$, thus recovering the pointwise formulation of Eq. 1.*

**Solving the Continuity Equation.**  The proof of the first part of Theorem 2 is a corollary of the following general statement.

**Theorem 5.** *Let $\Omega \subset \mathbb{R}^d$ be an open set and let $0 \leq t_0 < T$. Let $\boldsymbol{v} : [t_0, T] \times \Omega \to \mathbb{R}^d$ be a globally bounded velocity field that satisfies Eq. 20. Assume that the flow of $\boldsymbol{v}$ starting at time $t_0$, denoted by $\boldsymbol{X}$, is well defined in $[t_0, T] \times \Omega$ and that $\boldsymbol{X}_t : \Omega \to \Omega$ is a bilipschitz transformation of $\Omega$. Letting $\rho(\boldsymbol{x}, t)$ be defined by Eq. 17, then $\rho$ is a distributional solution to the continuity equation Eq. 1 according to Definition 1.*

We remark that, under the assumptions of Theorem 2, the flow map $\boldsymbol{X}$ is given and we build velocity field $\boldsymbol{v}$ that has $\boldsymbol{X}$ as a unique flow map. Hence, by construction, $\Omega$ is invariant under $\boldsymbol{X}_t$ for any $t \in [t_0, T]$. Thus, the assumptions of Theorem 5 are satisfied.

*Proof of Theorem 5.* By Theorem 3, the flow map $\boldsymbol{X}$ starting at time $t_0$ associated to $\boldsymbol{v}$ is well defined. Hence, defining $\rho$ by Eq. 17, for any $t \in [t_0, T]$ we have that $\rho(t, \cdot)$ is defined almost everywhere in $\mathbb{R}^d$. We check that $\rho$ is a distributional solution to Eq. 1 according to Definition 1. To begin, we check that $\rho \in L^\infty([t_0, T]; L^1(\mathbb{R}^d))$. Indeed, after the change of variables $\boldsymbol{X}_t^{-1}(\boldsymbol{x}) = \boldsymbol{y}$, using the Area formula with $dy = |\det J\boldsymbol{X}_t^{-1}(\boldsymbol{x})| dx$, we have that

$$\int_\Omega |\rho(t, \boldsymbol{x})| \, dx = \int_\Omega |\rho_{t_0}(\boldsymbol{X}_t^{-1}(\boldsymbol{x}))| |\det J\boldsymbol{X}_t^{-1}(\boldsymbol{x})| \, dx = \int_\Omega |\rho_{t_0}(\boldsymbol{y})| \, dy.$$

Therefore, the total mass is preserved along the time evolution. Then, fix a test function $\phi \in C_c^\infty([t_0, T) \times \Omega)$ and, performing again the change of variables $\boldsymbol{y} = \boldsymbol{X}_t^{-1}(\boldsymbol{x})$, we have that

$$\int_{t_0}^T \int_\Omega [\partial_t \phi + \boldsymbol{v} \cdot \nabla \phi] \rho \, dx \, dt$$

$$= \int_{t_0}^T \int_\Omega [\partial_t \phi(t, \boldsymbol{x}) + \boldsymbol{v}(t, \boldsymbol{x}) \cdot \nabla(t, \boldsymbol{x}) \phi] \rho_{t_0}(\boldsymbol{X}_t^{-1}(\boldsymbol{x})) |\det J\boldsymbol{X}_t^{-1}(\boldsymbol{x})| \, dx \, dt$$

$$= \int_{t_0}^T \int_\Omega [\partial_t \phi(t, \boldsymbol{X}_t(\boldsymbol{y})) + \boldsymbol{v}(t, \boldsymbol{X}_t(\boldsymbol{y})) \cdot \nabla \phi(t, \boldsymbol{X}_t(\boldsymbol{y}))] \rho_{t_0}(\boldsymbol{y}) \, dy \, dt.$$

Recalling that $X_t$ is the flow map generated by the velocity field $v$ starting at time $t_0$, the latter is equal to

$$\int_{t_0}^T \int_\Omega [\partial_t \phi(t, X_t(y)) + \partial_t X_t(y) \cdot \nabla \phi(t, X_t(y))] \rho_{t_0}(y) \, dy \, dt$$

$$= \int_\Omega \rho_{t_0}(y) \int_{t_0}^T \frac{d}{dt} \phi(t, X_t(y)) \, dt \, dy,$$

after using Fubini's Theorem and the chain rule for the derivatives. Thus, by the Fundamental Theorem of Calculus, we have that

$$\int_\Omega \rho_{t_0}(y) \int_{t_0}^T \frac{d}{dt} \phi(t, X_t(y)) \, dt \, dy = \int_\Omega \rho_{t_0}(y)[\phi(T, X_T(y)) - \phi(0, X_{t_0}(y))] \, dy$$

$$= -\int_\Omega \phi(t_0, y) \rho_{t_0}(y) \, dy,$$

since $\phi(T, \cdot) \equiv 0$ and $X_{t_0}$ is the identity map. $\qquad \square$

Finally, we are able to conclude the proof of Theorem 2.

*Conclusion of the proof of Theorem 2.* We discussed the fact that $\rho$ is a pointwise solution to the continuity equation Eq. 1 in Remark 2. Indeed, by the explicit formula Eq. 17 it is clear that $\rho$ is $C^\infty([t_0, T) \times \Omega)$ and that $\rho$ is nonnegative whenever $\rho_{t_0}$ is nonnegative. Thus, we check that Eq. 19 is satisfied. Indeed, by the chain rule we have that

$$\frac{d}{dt} \log(\rho(t, X_t(x))) = \frac{1}{\rho(t, X_t(x))} \left[ \partial_t \rho(t, X_t(x)) + \nabla \rho(t, X_t(x)) \cdot \frac{d}{dt} X_t(x) \right]$$

$$= \frac{1}{\rho(t, X_t(x))} [\partial_t \rho(t, X_t(x)) + \nabla \rho(t, X_t(x)) \cdot v(t, X_t(x))]$$

$$= \frac{1}{\rho(t, X_t(x))} \bigg[ \partial_t \rho(t, X_t(x)) + \nabla \cdot (v(t, X_t(x))\rho(t, X_t(x)))$$

$$- (\nabla \cdot v(t, X_t(x)))\rho(t, X_t(x)) \bigg]$$

$$= -\nabla \cdot v(t, X_t(x)),$$

since $\rho$ satisfies the continuity equation at any point $(t, x) \in (t_0, T) \times \Omega$. $\qquad \square$

## A.2 CALCULATING FOR THE DENSITY AND VELOCITY

In this subsection we explicitly provide the steps to get to Eq. 12 and Eq. 13.

### A.2.1 CALCULATING THE DENSITY

We now report explicitly the steps to get to Eq. 12.

Let $f$ and $g$ be diffeomorphisms on $\mathbb{R}^d$, and $A$ and $B$ positive definite matrices. We denote with $J(f)(g(x))$ the Jacobian of $f$ evaluated at the point $g(x)$. In the following part we make use of identities that follow from the chain rule, the inverse function theorem, and properties of the determinant:

$$J(f \circ g)(x) = J(f)(g(x)) \, J(g)(x), \tag{27}$$

$$|\det J(f)(x)| = \left| \det J(f^{-1})(f(x)) \right|^{-1}, \tag{28}$$

$$|\det(A \cdot B)| = |\det(A)| \cdot |\det(B)|. \tag{29}$$

$$\hat{\rho}(t, x) = \hat{\rho}_{t_0}(\hat{X}_t^{-1}(x))|\det J(\hat{X}_t^{-1})(x)| \tag{30}$$

$$= \hat{\rho}_{\text{base}}\left((\Phi_{t_0} \circ \hat{X}_t^{-1})(x)\right) \left|\det J(\Phi_{t_0})(\hat{X}_t^{-1}(x))\right| \left|\det J(\hat{X}_t^{-1})(x)\right| \tag{31}$$

$$\hat{\rho}_{\text{base}}\left((\Phi_{t_0} \circ \hat{\boldsymbol{X}}_t^{-1})(\boldsymbol{x})\right) = \hat{\rho}_{\text{base}}\left((\Phi_{t_0} \circ \Phi_{t_0}^{-1} \circ \Phi_t)(\boldsymbol{x})\right) = \hat{\rho}_{\text{base}}(\Phi_t(\boldsymbol{x})) \tag{32}$$

$$\left|\det J(\Phi_{t_0})(\hat{\boldsymbol{X}}_t^{-1}(\boldsymbol{x}))\right| = \left|\det J(\Phi_{t_0})\left((\Phi_{t_0}^{-1} \circ \Phi_t)(\boldsymbol{x})\right)\right| \tag{33}$$

$$= \left|\det J(\Phi_{t_0}^{-1})\left((\Phi_{t_0} \circ \Phi_{t_0}^{-1} \circ \Phi_t)(\boldsymbol{x})\right)\right|^{-1} \tag{34}$$

$$= \left|\det J(\Phi_{t_0}^{-1})(\Phi_t(\boldsymbol{x}))\right|^{-1} \tag{35}$$

$$\left|\det J(\hat{\boldsymbol{X}}_t^{-1})(\boldsymbol{x})\right| = \left|\det J(\Phi_{t_0}^{-1} \circ \Phi_t)(\boldsymbol{x})\right| \tag{36}$$

$$= \left|\det J(\Phi_{t_0}^{-1})(\Phi_t(\boldsymbol{x}))\right| \cdot \left|\det J(\Phi_t)(\boldsymbol{x})\right| \tag{37}$$

Combing the three terms we get:

$$\hat{\rho}(t, \boldsymbol{x}) = \hat{\rho}_{t_0}(\hat{\boldsymbol{X}}_t^{-1}(\boldsymbol{x}))|\det J(\hat{\boldsymbol{X}}_t^{-1})(x)| \tag{38}$$

$$= \hat{\rho}_{\text{base}}(\Phi_t(\boldsymbol{x})) \underbrace{\left|\det J(\Phi_{t_0}^{-1})(\Phi_t(\boldsymbol{x}))\right|^{-1} \cdot \left|\det J(\Phi_{t_0}^{-1})(\Phi_t(\boldsymbol{x}))\right|}_{=1} \cdot |\det J(\Phi_t)(\boldsymbol{x})| \tag{39}$$

$$= \hat{\rho}_{\text{base}}(\Phi_t(\boldsymbol{x})) |\det J(\Phi_t)(\boldsymbol{x})| \tag{40}$$

### A.2.2 CALCULATING THE VELOCITY WITHOUT INVERTING THE FLOW.

We now report explicitly the steps to get to Eq. 13. Firstly, we show that the velocity can be expressed in terms of the flow bijection $\Phi_t$ and its inverse $\Phi_t^{-1}$.

$$\hat{\boldsymbol{v}}(t, \boldsymbol{x}) = \frac{\partial \hat{\boldsymbol{X}}_t}{\partial t}\left(\hat{\boldsymbol{X}}_t^{-1}(\boldsymbol{x})\right) \tag{41}$$

$$= \frac{\partial(\Phi_t^{-1} \circ \Phi_{t_0})}{\partial t}\left((\Phi_{t_0}^{-1} \circ \Phi_t)(\boldsymbol{x})\right) \tag{42}$$

$$= \frac{\partial(\Phi_t^{-1})}{\partial t}\left(\underbrace{(\Phi_{t_0} \circ \Phi_{t_0}^{-1}}_{=1} \circ \Phi_t)(\boldsymbol{x})\right) \tag{43}$$

$$= \frac{\partial \Phi_t^{-1}}{\partial t}\left(\Phi_t(\boldsymbol{x})\right) \tag{44}$$

In a second step, the velocity can be written without the need for explicit inversion of the bijective layer $\Phi_t$, allowing for efficient computation in practice.

Let $\Phi_t$ and $\Phi_t^{-1}$ be the maps from $\boldsymbol{x}_t$ to $\boldsymbol{z}$ and vice versa respectively.

$$\Phi_t(\boldsymbol{x}_t) = \boldsymbol{z}, \quad \Phi_t^{-1}(\boldsymbol{z}) = \boldsymbol{x}_t \tag{45}$$

Clearly, $\Phi_t\left(\Phi_t^{-1}(\boldsymbol{z})\right) = \boldsymbol{z}$ and $\boldsymbol{z}$ does not depend on time, i.e. $\frac{d}{dt}\boldsymbol{z} = 0$. We can now explicitly compute the total derivative and find a formulation for the velocity that requires inverting a Jacobian instead of inverting the map $\Phi_t$.

$$\frac{d}{dt}\left(\Phi_t\left(\Phi_t^{-1}(\boldsymbol{z})\right)\right) = \frac{d}{dt}\boldsymbol{z} = 0 \tag{46}$$

$$\Rightarrow \frac{\partial \Phi_t}{\partial t}\left(\overbrace{\Phi_t^{-1}(\boldsymbol{z})}^{=\boldsymbol{x}_t}\right) + \overbrace{\frac{\partial \Phi_t}{\partial \boldsymbol{x}_t}\left(\Phi_t^{-1}(\boldsymbol{z})\right)}^{=J\Phi_t(\boldsymbol{x}_t)} \frac{\partial \Phi_t^{-1}}{\partial t}\left(\overbrace{\boldsymbol{z}}^{\Phi_t(\boldsymbol{x}_t)}\right) = 0 \tag{47}$$

$$\Rightarrow \frac{\partial \Phi_t^{-1}}{\partial t}\left(\Phi_t(\boldsymbol{x}_t)\right) = -[J\Phi_t(\boldsymbol{x}_t)]^{-1} \frac{\partial \Phi_t}{\partial t}(\boldsymbol{x}_t) \tag{48}$$

A.3 IMPLEMENTATION

This section provides high-level descriptions of the implementations of the models used in all experiments. For details on experiment-specific implementations, we refer to the corresponding sections A.4 to A.6.

**LFlows.** Our code for LFlows is based on the *nflows* library for bijective neural networks (Durkan et al., 2020). We extended *nflows* by providing a range of additional (conditional) transformations, such as invertible densenets, in addition to the modifications required for the LFlows. The code is provided as supplementary material.

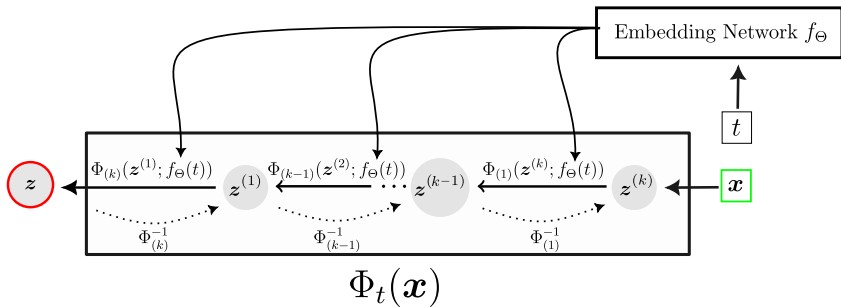

Figure A.6: General Architecture used for the conditional bijective layers based on embedding networks. Dotted lines indicate the inverse direction).

**SLDA.** We represent the density $\rho_{t_0}$ at the departure time $t_0$ with a bilinear interpolation of a learnable equi-distant mesh in 2D or 3D. We parameterize the velocity with a neural network with smooth activation functions. In 2D settings, we evaluate the divergence in Eq. 3 exactly via autograd, whereas we stochasticly estimate it in 3D during training (Grathwohl et al., 2019). To solve multiple ODEs with differing start times, in parallel we leverage the rescaling trick discussed in Appendix F of Chen et al. (2020). In all settings we use a Dormand-Prince solver of order 5 with absolute and relative tolerance of 1e-5. The adjoint is computed with the *torchdiffeq* library (Chen, 2018).

**DFNNs** We rely on the vector-based parameterization of DFNNs given in Section 7.1 of Richter-Powell et al. (2022), which ensures non-negative densities.

**PINNs.** We implement standard PINNs without additional resampling schemes or loss terms, and use either ReLu (Section 5.1) or sinusoidal activations (Sitzmann et al., 2020) (Section 5.3). The collocation points are resampled each training iteration with quasi-random Sobol sequences.

A.4 EXPERIMENT: SIMULATED FLUID FLOW

**Data Generation.** We generate the data by defining a Lagrangian flow map for an initial unnormalized Gaussian mixture density. The initial density of the simulated problem is based on a mixture of 4 independent Gaussians arranged around the origin with varying radii and a standard deviation of 0.1. We defined the density to be restricted to $\Omega = [-4, 4]^d$ with $(\rho \boldsymbol{v})|_{\delta\Omega} = 0$:

$$\rho_0(\boldsymbol{x}) = \begin{cases} \sum_{i=1}^4 \frac{1}{4} \mathcal{N}(\boldsymbol{\mu_i}, 0.1\boldsymbol{I}) & \text{if } \boldsymbol{x} \in (-4, 4)^d \\ 0 & \text{else} \end{cases} \tag{49}$$

The changes in density on the $xy$ axes are simulated by directly parameterizing the Lagrangian solution map $\boldsymbol{X}_t(\boldsymbol{x}_0) : (-4, 4)^2 \mapsto (-4, 4)^2$ for $t \in [0, 1.2]$:

$$\boldsymbol{X}_t(\boldsymbol{x}_0) = 4 \cdot \tanh\left(\frac{(0.5t + 1)}{4} A_{rot}(t) A_{scale}(t) \Big(4 \cdot \text{atanh}(0.25\boldsymbol{x}_0) + \text{shift}(t)\Big)\right) \tag{50}$$

with

$$A_{rot}(t) = \begin{bmatrix} \cos(2\pi t) & -\sin(2\pi t) \\ \sin(2\pi t) & \cos(2\pi t) \end{bmatrix} \tag{51}$$

$$A_{scale}(t) = \begin{bmatrix} 1 + 0.1t & 0 \\ 0 & 1 + 0.1t \end{bmatrix} \tag{52}$$

$$\text{shift}(t) = \sin(\pi t) \begin{bmatrix} 0.6 \\ -0.6 \end{bmatrix} \tag{53}$$

In 3D, the $z$ component of $\boldsymbol{X}_t$ is always an identity map, limiting the dynamics to the $xy$ axis.

The density and velocity are then given by:

$$\rho(t, \boldsymbol{x}) = \rho_0\Big(\boldsymbol{X}_t^{-1}(\boldsymbol{x})\Big)|\det J\boldsymbol{X}_t^{-1}(\boldsymbol{x})| \tag{54}$$

$$\boldsymbol{v}(t, \boldsymbol{x}) = \frac{\partial \boldsymbol{X}_t}{\partial t}(\boldsymbol{X}_t^{-1}(\boldsymbol{x})) \tag{55}$$

where all involved derivatives are computed using automatic differentiation. Observations are available at 21 equidistant timesteps in the range $[0, 1]$. The test data set covers the time range $[0, 1.2]$. To simulate noisy measurements, additional Gaussian noise is added to the observed velocities and log densities during training.

### A.4.1 TRAINING AND ARCHITECTURE DETAILS

**Hyperparameter Optimization.** We optimize each model based on the explained variance ($R^2$) of the density on validation data. Firstly, we manually performed a general architecture selection. Subsequently, we tuned parameters such as the number of layers, units, learning rate, loss- and regularization weights with the black-box optimization framework Optuna [3](Akiba et al., 2019), using the default Tree-structured Parzen Estimator as sampler. We trained the LFlows and PINNs on a minibatch size of 16384 and the SLDA on a minibatch size of 4096. As the 2nd order derivatives of the DFNNs require a lot of GPU memory, DFNNs were limited to a minibatch size of 2048. For the optimized hyperparameters of each model we refer to the provided code in the supplementary material.

**LFlows.** For the initial layer we first rescale the domain and then use a $atanh$ bijection for restricting the domain to $[-4, 4]^d$. For the remaining layers we used blocks consisting of invertible Dense Nets (i-DenseNet) (Perugachi-Diaz et al., 2021) followed SVD layer conditioned on the embedding. For the i-DenseNet we rely on sinusoidal activations $g(x) = \sin(15 * x)/15$. Each i-DenseNet has a depth of 3 and before each block we use an Activation Normalization layer. We enforce the Lipschitz constant of the i-DenseNet to be 0.97.

**DFNNs.** The Divergence-Free Neural Networks do not directly provide access to the velocity $\boldsymbol{v}$, but only to the flux $\boldsymbol{F} = (\rho\boldsymbol{v})$. Hence, calculating the velocity $\boldsymbol{v} = \boldsymbol{F}/\rho$ in low-density regions leads to numerical issues. To avoid this, we train DFNNs directly on the flux instead of the velocity. Furthermore, we require non-negative densities, so we use the parameterization with subharmonic functions discussed in Section 7.1 of Richter-Powell et al. (2022). As the predicted densities can still be zero, we train the DFNNs on the MSE of the densities (instead of log densities).

**PINNs.** To facilitate training of the PINN, we use sinusoidal activation functions as presented by (Sitzmann et al., 2020). We use a frequency multiplier of $\omega_0 = 12$ in the first layer. Collocation points are sampled within the full domain, uniformly distributed in $[0, 1.2] \times \Omega$. Instead of a purely random sampler, we rely on quasi-random low-discrepancy samples obtained via Sobol Sequences. In each minibatch, $2^{16}$ collocation points are sampled. The minimized PDE loss is $L(t, \boldsymbol{x}) = ||\partial_t\rho(t, \boldsymbol{x}) + \nabla \cdot (\rho(t, \boldsymbol{x})\boldsymbol{v}(t, \boldsymbol{x}))||^2$, averaged over the collocation points.

**SLDA** Relative and absolute tolerances of the solver during hyperparameter optimization were $10^{-3}$ and for the final run with the tuned hyperparameters $10^{-5}$. Lower tolerances during hyperparameter search were not feasible, as they significantly increased the runtime for the problem at

---

[3] https://optuna.org/

hand. For stable dynamics, the hypernetworks provided by the code of Grathwohl et al. (2019) were necessary. These hyper networks are conditioned on time and provide a network taking the space coordinates as input, i.e. $\boldsymbol{v}_{\Theta(t)}(\boldsymbol{x})$ with $\Theta$ being the hyper network. Using instead fully connected layers (that still fulfill the smoothness requirements) led in our experience to difficult dynamics for the adaptive ODE solvers.

**Boundary Conditions.** For PINNs, SLDA, and DFNNs the boundary condition $\rho(\boldsymbol{x})\boldsymbol{v}(\boldsymbol{x})|_{\delta\Omega} = \boldsymbol{0}$ is enforced via an additional penalty on points sampled at the boundary. For LFlows, the boundary was enforced via a bijection to $\Omega \setminus \delta\Omega$.

**Numerical Evaluation of Physical Consistency** We compare the predicted density $\hat{\rho}(t, \boldsymbol{x})$ with the numerical solution of the initial value problem uniquely defined by $\hat{\rho}(t_0, \boldsymbol{x})$, $\hat{\boldsymbol{v}}(t, \boldsymbol{x})$ and Eq. 3. We obtain the numerical solution with an 8th order Dormand-Prince ODE solver with an absolute and relative tolerance of 1e-5. We set $t_0 = 0$. For the locations where we evaluate the sMAPE, we consider 10 equidistant timesteps in $[0.1, 1.2]$. For each timestep, we first randomly sample locations in $(4, 4)^d$, and then we randomly subsample 2500 location with groundtruth densities larger than a threshold of 0.1 in 2D or 0.01 in 3D. With this procedure we avoid regions with near-zero density.

**Total Mass Regularization.** We penalize the total mass of the system for all methods except PINNs. For the LFlows and SLDA, we penalize the learnable normalization constant. For the DFNNs, no equivalent to the normalization constant is available. We instead introduce the penalty at points sampled on the domain $[0, 1.2] \times \Omega$. For PINNs additional regularization is not necessary. This is due to the side effect of the PDE loss being numerically small for small densities, leading to an automatic built-in penalty for large total mass.

**Computational Resources.** Each individual experiment for the synthetic data was run on individual NVIDIA TITAN X GPUs (12GB VRAM), using 20 CPU cores and 20GB RAM. To speed up hyperparameter tuning, up to 8 experiments were run in parallel using a SLURM-based compute cluster.

## A.5 EXPERIMENT: DYNAMICAL OPTIMAL TRANSPORT

In Figure A.5 and Figure A.5 interpolated densities are shown for the 8Gaussians↔Circles dataset and the Pinwheel↔8Gaussians dataset.

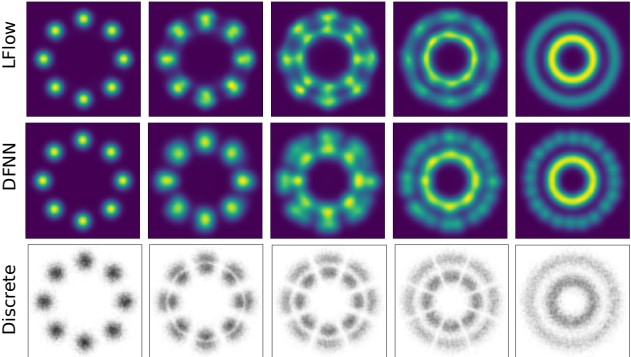

Figure A.7: Approximations of the 2D optimal transport map for the 8Gaussians↔Circles dataset with LFlows, DFNNs, and a discrete reference.

### A.5.1 EVALUATING THE INTEGRAL

For evaluating the integral required for the objective in Eq. 14, we reuse the code and method provided by Richter-Powell et al. (2022). The integral is estimated via importance sampling, with the sampling distribution $q(t, \boldsymbol{x}) = q(t)q(\boldsymbol{x})$. Samples in time are drawn uniformly with $q(t) \sim$

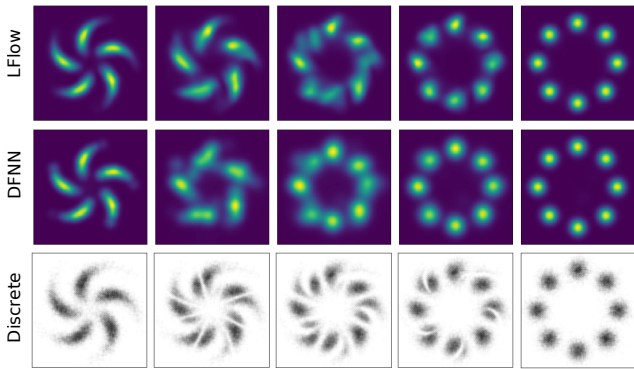

Figure A.8: Approximations of the 2D optimal transport map for the Pinwheel↔8Gaussians dataset with LFlows, DFNNs, and a discrete reference.

$\mathcal{U}(t_0, t_1)$. Samples in space are drawn from a uniform mixture

$$q(\boldsymbol{x}) = \frac{1}{3}p_{t_0}(\boldsymbol{x}) + \frac{1}{3}p_{t_1}(\boldsymbol{x}) + \frac{1}{3}\mathcal{U}_\Omega(\boldsymbol{x}) \tag{56}$$

with $\mathcal{U}_\Omega$ being the uniform distribution on the domain.

### A.5.2 DISCRETE ESTIMATE OF W2

For estimating the $W_2^2$ distance with a discrete reference method we rely on 50000 samples. In our setting, more samples lead to convergence issues of the numerical solver.

As the discrete $W_2^2$ estimate depends on the number of samples $n$, we explore an increasing number of samples and repeated runs. For each fixed $n$ we repeat the estimate 10 times with different seeds and visualize the resulting box plots in A.5.2. We note that the differences with increasing sample size are comparatively small relative to the differences between the continuous methods in Section 5.2.

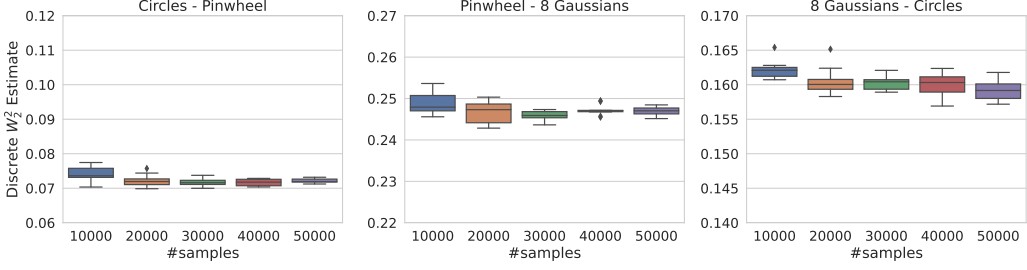

Figure A.9: Change of the discrete $W_2^2$ estimate based on increasing number of samples. Each boxplot is based on 10 repeated runs with different seeds.

### A.5.3 IMPLEMENTATION

**LFlows.** For LFlows we use 10 blocks of i-DenseNets, each preceded by an Activation Normalization layer. Each i-DenseNet has a depth of 5 with sinusoidal activations $g(x) = \sin(15 * x)/15$. We enforce the Lipschitz constant of the i-DenseNet to be 0.97. For the embedding of the time condition we use a residual neural network with 1 hidden layer, swish activations, and 128 hidden features. The dimension of the time embedding, i.e. the output of the embedding network, is of dimension 10. The model was trained with the ADAM optimizer for 5000 iterations with a learning rate of 2e-3 and 2048 data points per iteration.

**DFNNs.** For DFNNs we directly used the experiment code provided by Richter-Powell et al. (2022). That is, the DFNNS are based on the parameterization with subharmonic functions discussed in Section 7.1 of Richter-Powell et al. (2022). The 128 mixtures of the subharmonic function are parameterized with a 4 layer neural network with 96 hidden features and swish activations. A fixed $\lambda$ weight of 50 is set for the OT-penalty for all datasets. The DFNN is trained with the ADAM optimizer and a learning rate of 1e-3 over 10 000 iterations with 256 data points per iteration.

**Computational Resources.** The experiment was run on individual NVIDIA TITAN X GPUs (12GB VRAM), using 20 CPU cores and 20GB RAM.

## A.6 EXPERIMENT: BIRD MIGRATION

**About the data.** The data provided by Nussbaumer et al. (2021) is originally based on weather radar measurements made available by the *European Operational Program for Exchange of Weather Radar Information* (EUMETNET/OPERA). The vertical profiles, i.e. density and velocity estimates at different altitude levels, were provided by the *European Network for the Radar surveillance of Animal Movement* (ENRAM), based on *vol2bird*[4], an algorithm for preprocessing raw radar scans. The raw data consists of volume scans from Doppler radars, measuring reflectivity and radial velocity of the surroundings. By filtering out biological and environmental scatters, it is possible to retain scans that mostly contain bird movements. Based on the reflectivity and radial velocity, the average bird density and velocity within a 15km radius is estimated for multiple altitude bins. For details about this process, we refer to Dokter et al. (2011). The final density and velocity measurements we use are openly available[5].

**Preprocessing.** The positions of the radars are given in the WGS84 coordinate reference system. We project it to the Cartesian reference system ETRS89-extended (EPSG:3035), effectively projecting longitude/latitude to $x$ and $y$ coordinates given in meters. As an additional preprocessing step, we excluded velocities that were measured together with a near-zero density. To generate the data set used in our experiment, we directly concatenated multiple nights and remove the daytime during which no measurements are available. In the supplementary material we provide code for downloading and preprocessing the data.

**Numerical Evaluation of Physical Consistency** We compare the predicted density $\hat{\rho}(t, \boldsymbol{x})$ with the numerical solution of the initial value problem uniquely defined by $\hat{\rho}(t_0, \boldsymbol{x})$, $\hat{\boldsymbol{v}}(t, \boldsymbol{x})$ and Eq. 3. The time of the earliest available datapoint within the three nights serves as $t_0$. We evaluated the error at a spatial equidistant grid of sidelength 50 in the $xy$ dimension. The $z$ coordinate is randomly sampled for each of the grid entries. We evaluate the consistency loss at these $xyz$ coordinates 10 time steps between the start and end of the three selected nights, and average the sMAPE over all $xyzt$ coordinates. For all models we use a 8th order Dormand-Prince ODE solver with an absolute and relative tolerance of 1e-5, which is one order higher than the ODE solver used for SLDA. The MLP has no $z$-component of the velocity, as no measurements of it are in the data. We thus set this component of the MLP to zero for computing the numerical ODE solution. The other models indirectly learn a $z$-component by enforcing the CE in 3D.

### A.6.1 TRAINING AND ARCHITECTURE DETAILS

**LFlows.** As first layer of the LFlow we use a $atanh$ bijection that constrains $\Omega$ to a rectangular volume that is multiple times larger than the spatial extent of the radar positions. The following layers consist of 10 blocks of invertible Dense Nets (i-DenseNet) (Perugachi-Diaz et al., 2021), where we instead use sinusoidal activations $g(x) = \sin(10 * x)/10$. Each block has a depth of 5 and before each block we use an Activation Normalization layer. We enforce the Lipschitz constant of the i-DenseNet to be $0.97$. We use a 2 layer residual network with a width of 128 for embedding the time before passing it to the i-DenseNet.

We initialized the log of the normalization constant $\ln(c)$ with $18.2$ and set the weight of the total mass penalty to 1e-3. We trained for 50 epochs with a minibatch size of 16384 using the ADAM

---

[4]`https://github.com/adokter/vol2bird`
[5]`https://zenodo.org/record/4587338/`

optimizer with a learning rate of 1e-2, a weight decay of 2e-3 and a cosine annealing learning rate schedule.

**SLDA.** For the SLDA we represent the initial density with a grid of size $20 \times 20 \times 20$. Values in between mesh points are evaluated with bilinear interpolation. We parameterize the velocity network with a hypernetwork as provided by the code of Grathwohl et al. (2019), similar to the simulated fluid flow experiment. The network consists of 5 layers with 512 hidden features and swish activations. We trained for 300 epochs with a minibatch size of 16384 using the ADAM optimizer with a learning rate of 1e-3, a weight decay of 5e-3 and a cosine annealing learning rate schedule.

**DFNN.** For the DFNN we use the parameterization for non-negative densities based on subharmonic functions (Section 7.1 Richter-Powell et al. (2022)). The model consists of 4 layers with 256 hidden features and Swish activations, which parameterize 64 mixture components for the subharmonic function. We used a minibatch size of 2048 due to memory constraints. The model was trained for 100 epochs with a cosine annealing learning rate schedule.

**MLP.** We trained a 10 layer residual neural network with ReLu activations, intermediate batch normalization and 256 hidden features. The model was trained for 100 epochs with a minibatch size of 16384 using the ADAM optimizer with a learning rate of 1e-3, a weight decay of 1e-3 and a cosine annealing learning rate schedule.

**PINN.** For the PINN we use the same architecture as for the MLP, but with an additional PDE loss. We use 100 000 collocation points sampled from a quasi-random Sobol sequence and weight the PDE loss with 7e-4. The PINN was trained for 300 epochs with a minibatch size of 16384 using the ADAM optimizer with a learning rate of 1e-2, a weight decay of 1e-2, and a cosine annealing learning rate schedule

**Computational Ressources.** The experiment was run on an A100 GPU (40GB VRAM), using 20 CPUs and 30GB RAM. Repeated experiments were parallelized on a SLURM-based compute cluster.

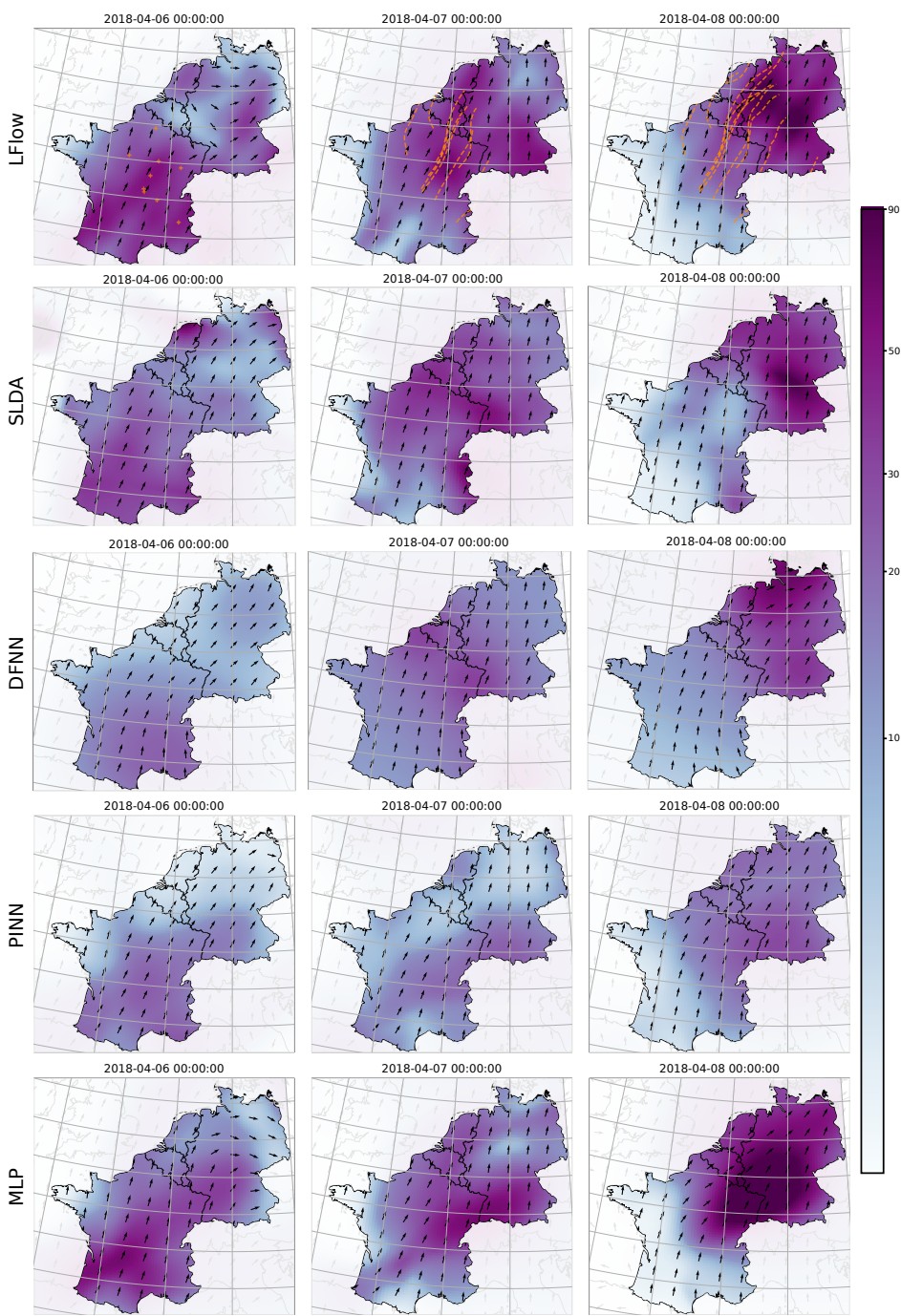

Figure A.10: Predicted density and normalized velocity integrated over z-axis for all considered models.

### A.6.3 Total Mass Penalty

To motivate the total mass penalty we showcase its effect on bird migration predictions using LFlows with varying penalty weights. The different predictions are shown in Figure A.11

Without any penalty, the network can freely explain the training data with large amounts of total mass. This leads to significant and never observed densities outside of the observed region. However, with an active total mass penalty effectively being a zero prior the network is instead encouraged to decrease the total mass. The total mass penalty weight allows to explore this aspect of the ill-posedness of the problem.

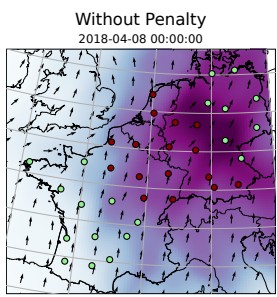 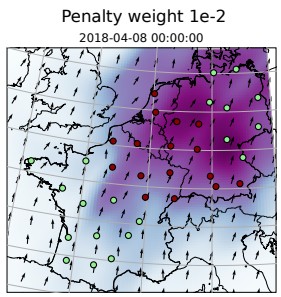 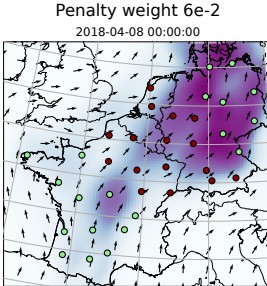

Figure A.11: Predictions of the LFlow trained without *(left)* and with (*right*) total mass penalty. While predictions at observed radar stations barely change, the total mass outside of the observed region is significantly reduced.

The variation in predictions can be further summarized in a single visualization. Similar to before, we train different models for varying total mass penalty weights. We calculate the relative standard deviation of the predicted flux (i.e. the product of density and velocity) at each spatial location for a fixed time. Areas with higher relative standard deviations correspond to areas with largely varying explanations. Figure A.12 shows the resulting map for a single time frame for the bird migration setting. The areas which are never observed and are completely dependent on a prior show the highest variation. Areas closer to the train radar stations (light green points) have a lower variation.

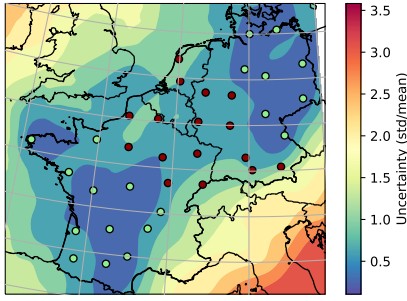

Figure A.12: Relative standard deviation of predicted flux with varying mass penalties.

