# OpenReview forum: "Lagrangian Flow Networks for Conservation Laws"
_ICLR.cc/2024/Conference — ICLR 2024 spotlight_

### Official Review · Reviewer_RQ15 · 2023-10-24

**Soundness:** 3 good
**Presentation:** 4 excellent
**Contribution:** 2 fair
**Rating:** 6
**Confidence:** 4

**Summary:**

This paper proposes LFlow, a neural parameterization of both the density and the velocity field that adhere to the continuity equation. This is achieved with a link between time-conditioned diffeomorphisms and Lagrangian solution maps for the continuity equation. Two application settings were considered, where in one setting the user has sparse data on both density and velocity, and in another setting, the user has no data but knows the equations. Low-dimensional experiments are done to demonstrate the effectiveness of the proposed parameterization.

**Strengths:**

* The writing of the paper is clear.
* Both settings are interesting and the proposed method has been shown to outperform the alternatives (PINN, DFNN, SLDA).

**Weaknesses:**

* To my knowledge, the idea of using the Lagrangian view of the continuity equation to build a neural parameterization that gives both velocity and density access is not new.
  * In Section 3.1 and 3.2 of [1], the TIPF parameterization appears to be identical to LFlow proposed in the present paper, with the only difference being that in (13) of the present paper, $\Phi_t$ appears instead of the inverse, which is the case of TIPF.
  * This is a minor difference for TIPF, since in [1] the conditional normalizing flow used is RealNVP [2], which has ready access to the inverse flow map.
  * Could the authors elaborate on the advantages of avoiding the inverse here?
  * Could the authors also comment on the choice of the Lipschitz-constrained invertible densenets as the backbone architecture? What is the advantage/disadvantage compared to e.g. RealNVP [2]?
* The experiments section is weak in my opinion, and weaker than that in [Richter-Powell 2022]. If I understand correctly, all experiments are in 2D. How would the parameterization fare in higher dimensions? Higher dimensional experiments are done in [Richter-Powell 2022]. In [1] it is found that TIPF does not work well in dimension >= 10. I suspect the same behavior occurs here, due to the reason that invertible pushforward networks might not be expressive (or might be hard to train).

[1] Li, Lingxiao, Samuel Hurault, and Justin Solomon. "Self-consistent velocity matching of probability flows." arXiv preprint arXiv:2301.13737 (2023).

[2] Dinh, Laurent, Jascha Sohl-Dickstein, and Samy Bengio. "Density estimation using real nvp." arXiv preprint arXiv:1605.08803 (2016).

**Questions:**

1. At the bottom of the first page, it reads "in setting (ii) we measure only the density of the fluid but we know additional equations defining the velocity." If density is already known for all time, then isn't it already enough to reconstruct the velocity without needing to know any additional equations, by simply solving the continuity equation?
2. In the evaluation in Section 5.2, the gold standard of $W_2^2$ is computed using discrete estimates (and then averaged). However, the discrete estimate of W2 is known to be biased. How can one be sure that the red bars on the right of Figure 3 are accurate? Would the bar shift if more samples were used?
3. In (15), how is the double integral computed in practice? It looks like it requires uniform samples from the domain $\Omega$, which could be very ineffective if the flow only concentrates on a small portion of the domain.

---

> ### Author Response · Authors · 2023-11-13
> **i) Comparison to TIPF**
>
> We are thankful to the reviewer for pointing out a related work, which we will include in the revised version of the paper. However, we respectfully disagree that this constitutes lack of novelty.
>
> The proposed approach shares only a high-level similarity with TIPF. Both papers use known results from classical theory, namely that conditional diffeomorphisms define probability densities that satisfy the continuity equation by construction. However, (i) the class of problems addressed by the two papers is fundamentally different and (ii) the proposed architectures are consequently designed for different purposes.
>
> &nbsp;
>
> ### (i) Scope of the papers:
> - TIPF consider well-posed forward problems for probability flows, where initial conditions, boundary conditions, and the full system of equations are known. Initial conditions are evolved forward in time with conditional bijections that fulfill the continuity equation by construction, providing an advantage compared to numerical methods.\
>
> &nbsp;
>
> - In contrast, we consider inverse problems for real physical phenomena. We rely only on sparse (and noisy) observations, with unknown initial conditions, velocity and total mass. This is a challenging ill-posed setting and cannot be directly approached with classical numerical forward solvers.
>
> &nbsp;
> ### (ii) Proposed architecture:
> - TIPF evolve a known initial density $p_{t_0}$ over time by directly parameterizing the Lagrangian flow map, which we denote as $X_t$ in Section 3.1. We shortly discuss this parameterization in Section 3.2. As $X_t$ is directly parameterized, $X_{t_0}$ must be enforced to be the identity map with an additional loss. Since there is no base density, observations are always mapped back to the known initial density $p_{t_0}$. One important limitation is that the bijection has to be inverted, which is computationally expensive. Therefore, TIPF rely on bijections with analytical inverses, which however severely limit the expressivity of the model.
>
> &nbsp;
> - In contrast, LFlows do **not** directly parameterize $X_t$ and $\rho_{t_0}$. Instead, only a single time-conditioned bijection $\Phi_t$ is learned, such that $X_t(x) = \Phi_t^{-1}(\Phi_{t_0}(x))$ (see Section 3.2). By mapping the time dependent density to a common base density, we do not even need an explicit $\rho_{t_0}$. Importantly, our approach enables us to avoid computing the inverse explicitly. We can compute the velocity by simply inverting the Jacobian of $\Phi_t$ instead of explicitly inverting $\Phi_t$. We would like to highlight that this is a significant contribution, since unlike TIPS we are free to choose highly flexible bijections with unknown or expensive inverses.
>
> &nbsp;
>
> ## Questions
> We first address questions and concerns raised in the "weaknesses" sections. We will address the remaining questions in a separate post.
>
> >Could the authors elaborate on the advantages of avoiding the inverse here?
>
> Highly expressive bijective layers usually result in the loss of an efficient (analytical) inverse. Lipschitz constrained residual networks for example require a fixed-point iteration for obtaining the inverse. Backpropagating through such an operation (for minimizing a velocity loss) quickly turns infeasible for moderately sized networks. By completely avoiding the inverse we do not share the same limitations for the choice of the bijective layers as TIPF.
>
> >Could the authors also comment on the choice of the Lipschitz-constrained invertible densenets as the backbone architecture? What is the advantage/disadvantage compared to e.g. RealNVP [2]?
>
> Invertible densenets have free-form Jacobians and showed high performance on generative modeling tasks compared to other models [3], including RealNVP. We further found them easy to finetune and less prone to training instabilities and overfitting compared competing bijective layers.
>
> >If I understand correctly, all experiments are in 2D. How would the parameterization fare in higher dimensions?
>
> We would like to correct the misunderstanding regarding the dimension. Both the  experiments in section 5.1 and the real-world bird migration experiment in 5.3 are in 3D. In the simulated fluid experiment we also show that the proposed LFlow outperforms competing methods by an even greater margin when going from 2D to 3D (see Figure 2, top right)
>
> >The experiments section is weak in my opinion, and weaker than that in [Richter-Powell 2022].
>
> Concerning the experiments, we respectfully disagree that our section is weaker compared to [Richter-Powell 2022]. From our point of view, the main purpose of PDE-constrained neural networks is the seamless incorporation of measurements for solving ill-posed inverse problems. Different to [Richter-Powell 2022], we actually evaluate such a setting with the bird migration experiment.
>
> [3] Perugachi-Diaz, et al. Invertible densenets with concatenated LipSwish. NeurIPS 2021.
>
> &nbsp;
>
> Please let us know if there are any raised concerns we missed to address.

---

> > ### Author Response · Authors · 2023-11-14
> > **(ii) Questions**
> >
> > ## Questions ("Weaknesses" Section)
> >
> > > How would the parameterization fare in higher dimensions? [...]  In [1] it is found that TIPF does not work well in dimension >= 10. I suspect the same behavior occurs here, due to the reason that invertible pushforward networks might not be expressive (or might be hard to train)
> >
> >
> > We assume "working well in dimensions >= 10" refers to (i) computational scalability, (ii) generalization capability, (iii) the enforcement of additional PDEs, and (iv) expressivity of the bijections.
> >
> > (i) In our 2D and 3D experiments we showed that the proposed LFlows scale better than competing methods in terms of memory and/or training time.
> >
> > (ii) While the generalization capability in higher dimensions might be interesting to study, we are not aware of any real world settings where a density or velocity of a fluid in high (>3) dimensional spaces is observed. The work of [1] considers probability flows in high dimensions rather than physical observations of densities.
> >
> > (iii) Settings like the ones discussed in [1] require enforcement of additional equations in high-dimensional spaces, similar to our optimal transport experiment. As in PINNs, a PDE-Loss has to be minimized on the support of the density by sampling collocation points in high dimensions. We agree that the proposed LFLows would suffer from the same limitations in high dimensions. The fulfillment of the continuity equation however remains unaffected by higher dimensions.
> >
> > (iv) Regarding expressivity, Normalizing Flows (and i-DenseNets) have already been succesfully employed in very high-dimensional settings (32x32 images) [3], leading to our assumption that the expressivity should not be the limiting factor.
> >
> >
> > > [...] Higher dimensional experiments are done in [Richter-Powell 2022]. [...]
> >
> > We note that the high-dimension (>>3) experiments of [Richter-Powell 2022] are about decomposing vector fields into irrotational and divergence-free components. This is specific to divergence-free networks and unrelated to our work.
> >
> >
> >
> > &nbsp;
> >
> > ----------
> > ## "Questions" Section
> >
> > > 1. [...] it reads "in setting (ii) we measure only the density of the fluid but we know additional equations defining the velocity." If density is already known for all time, then isn't it already enough to reconstruct the velocity without needing to know any additional equations, by simply solving the continuity equation?
> >
> > &nbsp;
> >
> > We never assume dense observations at all times, but we agree that in such a setting one could and should directly solve the continuity equation numerically. In contrast, for our setting we always assume sparse measurements in spacetime. We see how this is not apparent from the quoted sentence. We will state it more clearly in the revised version of the paper.
> >
> > &nbsp;
> >
> > --------------------
> > > 2. In the evaluation in Section 5.2, the gold standard of
> >  is computed using discrete estimates (and then averaged). However, the discrete estimate of W2 is known to be biased. How can one be sure that the red bars on the right of Figure 3 are accurate? Would the bar shift if more samples were used?
> >
> > We thank the reviewer for highlighting that, as we were unaware of this issue.
> > We will rerun the discrete solvers with up to 50.000 samples (previously 20.000) to explore the behaviour and add the results to the appendix in a revised version. For the Circle-Pinwheel example, we can already state that the estimate remains rather stable, going from 7,29E-02 (+- 2,10E-03) to 7,26E-02 (+- 6,66E-04), where we report AVG (+- STDEV).
> >
> >
> > &nbsp;
> >
> > --------------------
> > > 3. In (15), how is the double integral computed in practice? It looks like it requires uniform samples from the domain, which could be very ineffective if the flow only concentrates on a small portion of the domain.
> >
> > For the synthetic experiment we adopted the procedure of [Richter-Powell 2022] to be consistent with their setting. They rely on importance sampling for evaluating the integral (15), with a sampling distribution $q(t,x)$. Samples in time are drawn uniformly, i.e. $q(t)\sim U(t_0, t_1)$. Samples in space are drawn from a uniform mixture: $q(x) = \frac{1}{3} p_{t_0}(x) + \frac{1}{3} p_{t_1}(x) + \frac{1}{3}U_\Omega$ with $U_\Omega$ being the uniform distribution on the domain.
> >
> > Different to [Richter-Powell 2022] we could also directly sample from the density, reformulating the integral as an expectation. To avoid backpropagating through the numerical inverse, one might also detach the density samples from the computational graph and use them as a proposal distribution for importance sampling.

---

> > > ### Comment · Reviewer_RQ15 · 2023-11-18
> > > **Reply to authors**
> > >
> > > Thank you for your thorough rebuttal and revision. I have read it closely and I agree a few of my critiques were not accurate. Thanks for the explanation. As such, I have increased my score.
> > >
> > > Here are some more comments.
> > > * The revised discussion of TIPF in the related work section is adequate. Previously I was concerned about a few overclaiming statements in the original submission. I think those statements have been revised.
> > > * While I agree the scope of this present paper is different than that of [1] in the (i) scenario of having data but without equations, I found the novelty aspect limited since the idea of using the same parameterization to automatically satisfy CE is not entirely new, and PINN is originally proposed to handle this scenario using the same idea of regressing the parameterized flow to observation data.
> > >   - I think it can be made more explicit in Section 3.2 of the difference between the proposed parameterization and that of [1]. To me, the two main differences are 1) LFlows avoids inverting the flow map Phi and instead inverts a Jacobian matrix, and 2) LFlow uses a flow map that pushes from a base reference distribution rather than an initial distribution.
> > > * In terms of the exhaustiveness of experiments compared to [Richter-Powell 2022], I now agree the current set of experiments is more practical due to the bird migration experiment.

---

### Official Review · Reviewer_fhDc · 2023-10-26

**Soundness:** 3 good
**Presentation:** 3 good
**Contribution:** 3 good
**Rating:** 8
**Confidence:** 3

**Summary:**

This paper expresses the continuity equation into the time-dependent density transformation to model the fluid densities and velocities. By introducing differentiable and invertible maps, the proposed LFlows can ensure the continuity equation inherently. The authors provide complete proof to support their insights. Typical experiments are also included. Experimentally, LFlows performs best in both performance and efficiency.

**Strengths:**

1.	The paper starts from an interesting insight. The proposed method is reasonable and well-supported.

2.	The authors conduct extensive experiments, covering both simulated and real-world dataset.

3.	The propose LFlows performs well in both performance and efficiency.

**Weaknesses:**

1.	About the presentation.

For me, the writings in Section 3.1 and 4 are confusing and hard-to-read.

In Section 3.1, the authors use several theorems in an informal way and sometimes not give the exact formalization of theorem. I prefer to place the definitions of theorems in the main text and leave the proof in appendix, such as Theorem 1,2 and the existence and uniqueness of Lagrangian flows for smooth vector fields.

In Section 4, I don’t think the implementation of LFlows are well described. The model architecture in Appendix A.3 should be placed in main text. Besides, the implementation for baselines should be deferred into the Appendix.

2.	How to adopt LFlows to process the spatially sparse data? More details are expected.

3.	About the noise observations in the real-world dataset. Since LFlows enable the strict continuity equation, does it come across problems when the input data is noisy even wrong? Some discussions are expected.

**Questions:**

All the questions are listed above.

---

> ### Author Response · Authors · 2023-11-15
>
> We thank the reviewer for the valuable feedback on the presentation of our work. We are eager to make the paper clearer and more accessible.
>
> --------
> &nbsp;
>
> > For me, the writings in Section 3.1 [...] are confusing and hard-to-read.
> > In Section 3.1, the authors use several theorems in an informal way and sometimes not give the exact formalization of theorem. I prefer to place the definitions of theorems in the main text and leave the proof in appendix, such as Theorem 1,2 and the existence and uniqueness of Lagrangian flows for smooth vector fields.
>
> We understand the argument with respect to the exact formulation of the theorems, and would generally agree with this preference.
>
> However, in a previous submission, we did provide exact formalizations and statements of the theorems in 3.1, which we now moved to A.1.1. This led to large misunderstandings among the reviews. We received the unanimous feedback that including the formal and large theorems in the main text decreased the clarity and potentially led the reader to believe that it was the main constribution of the work, instead of the starting point.
>
> We would thus lean towards keeping the current version of Section 3.1, with the full theorems provided in the Appendix A.1.
>
>
> -----------------
> &nbsp;
>
>
> > [...] the writings in Section [...] 4 are confusing and hard-to-read.
> > In Section 4, I don’t think the implementation of LFlows are well described. The model architecture in Appendix A.3 should be placed in main text. Besides, the implementation for baselines should be deferred into the Appendix.
>
> We thank the reviewer for highlighting these clarity issues, and agree that Section 4 can be improved by following these recommendations.
>
> We rewrote the implementation section accordingly. That is, Section 4 now only discusses the implementation of LFlows, and does so in a clearer way. The implementation of the baselines were deferred to the Appendix A.3.
>
> -----------------
> &nbsp;
>
> > How to adopt LFlows to process the spatially sparse data? More details are expected.
>
> LFlows were directly designed for sparse data and do not need any adaption. Since LFlows provide continuous predictions in space and time and do not rely on a grid, we can predict the density and velocity at any point in space and time. To avoid overfitting in very sparse data regimes, we recommend additional regularization to limit the capacity of LFlows, such as the weight decay we employ for the Bird Migration.
>
> All the experiments cover data regimes with some level of sparsity, be that in space or time. The synthetic 2D/3D experiments showcase continuous observations but only for parts of the domain. A truly sparse setting with irregularly spaced observations is explored with the bird migration experiment where 37 radar stations are the only information for three countries. Due to the nature of the Radar-Ornithology preprocessing pipeline, we only obtain a single density and velocity measurement for each radar at each time. This value does correspond to an average over a larger radius, but we never have access to the full radar volume scans. We refer to [1] for details on the radar data preprocessing.
>
> [1] Dokter, Adriaan M., et al. "Bird migration flight altitudes studied by a network of operational weather radars." Journal of the Royal Society Interface 8.54 (2011): 30-43.
>
>
> --------------
> &nbsp;
>
> > About the noise observations in the real-world dataset. Since LFlows enable the strict continuity equation, does it come across problems when the input data is noisy even wrong? Some discussions are expected.
>
> We do not observe issues with respect to noisy observations. We think such noise should be addressed in the loss function / likelihood. For example, by minimizing the MSE on the log1p transformed density we assume log(1p)-Gaussian noise. The problem of designing a problem specific noise model is however not unique to our method, and is an essential part of many classical machine learning and data assimilation problems.
>
> Another point is that we are in highly ill-posed settings. From the data alone it can not be determined whether mass is strictly conserved and comes from an unobserved region, or whether there is a sink or source. Thus, we argue that the motivation for enforcing a physical constraint in such a setting must come from domain knowledge. We assume that for few consecutive nights during migration, there is a negligible change in bird-mass compared to the resolution and accuracy of our sensors. If this assumption is wrong, then of course the strict enforcement of the constraint will be unhelpful and even harming for the prediction.

---

> ### Comment · Reviewer_fhDc · 2023-11-21
> **Thanks for your response and raise the score**
>
> I would like to thank the authors' efforts in clarifying my concerns. Considering the overall quality and experiments of this paper, I would like to raise my score to 8.

---

### Official Review · Reviewer_1yfJ · 2023-10-30

**Soundness:** 4 excellent
**Presentation:** 4 excellent
**Contribution:** 4 excellent
**Rating:** 8
**Confidence:** 3

**Summary:**

The paper introduces a neural network-based model designed to adhere to the continuity equation, even when dealing with scenarios where precise boundary and initial conditions are unknown. The core technical foundation relies on the classical theory of Lagrangian flows, providing a toolbox that ensures the evolution of density and velocities complies with the continuity equation.

**Strengths:**

The paper significantly contributes to the field of neural PDE methods by focusing on enforcing constraints. It marries neural networks with conditioning normalization flows, resulting in LFlows that satisfy the continuity equation by design.

The method is validated across various 2D and 3D problems, including a real-world bird migration dataset, which could potentially serve as a standard benchmark dataset for similar problems. Unlike previous methods that often require known PDEs, initial, and boundary conditions, this approach tackles data assimilation-style problems where many system parameters are unknown. This kind of problem could be particularly useful for biomedical imaging applications.

The paper rigorously compares its approach to Richter-Powell's divergence-free neural networks, showcasing the computational efficiency of its method without the need for higher-order autodiff. Additionally, it compares favorably against other standard baselines, such as PINNs, especially in spatially sparse settings.

**Weaknesses:**

The proposed method inherits limitations associated with normalizing flows and bijective layers, such as difficulties in handling discontinuities.

**Questions:**

Could the authors comment on the challenges they anticipate when scaling this approach to larger problems, such as modeling large turbulent flows in aerodynamics? How might the complexity of LFlows impact scalability in such scenarios?

How does the computational complexity of LFlows compare to existing methods for solving hydrodynamic flow problems, especially in high-dimensional scenarios? Are there specific trade-offs or advantages in terms of computational efficiency?

Does the methodology proposed in this paper have the potential to be generalized to tackle other partial differential equations (PDEs) beyond the continuity equation? If so, what challenges or modifications might be necessary when applying it to different PDEs?

---

> ### Author Response · Authors · 2023-11-14
>
> We thank the reviewer for the interest in our work and share their enthusiasm.
>
> &nbsp;
> -----
> ### Question 1
> > Could the authors comment on the challenges they anticipate when scaling this approach to larger problems, such as modeling large turbulent flows in aerodynamics? How might the complexity of LFlows impact scalability in such scenarios?
>
> We estimate that large-scale turbulent flows are currently outside the scope of LFlows. Our current applications mostly consider rather smooth spatiotemporal interpolations, while turbulent flows consist of complex and chaotic dynamics on multiple scales. Nonetheless, to approach such complex dynamics with LFlows, we would recommend to use neural spline transformations [0] as bijective layers. These are extremely flexible and provide localized (bin-wise) transformations, but very easily overfit in our settings with sparse measurements.
>
> In future work we plan to upscale the bird migration setting to longer time scales and consider covariates/hyperparameters influencing the density movement. This includes e.g. wind and rain data changing the migration behavior. In this upscaled setting we plan to condition each individual night on available covariates via the embedding networks, effectively splitting the problem into many (coupled) intra-night dynamics. Challenges mostly consider the design of the Embedding networks, as we currently rely on very simple architectures (MLPs with skip connections).
>
> [0] Durkan, Conor, et al. "Neural spline flows." Advances in neural information processing systems 32 (2019).
>
> &nbsp;
> -----
> ### Question 2
> >How does the computational complexity of LFlows compare to existing methods for solving hydrodynamic flow problems, especially in high-dimensional scenarios? Are there specific trade-offs or advantages in terms of computational efficiency?
>
> As we compared our work to multiple data assimilation methods, we assume this question refers to solving forward problems. LFlows were however mainly designed with ill-posed data-assimilation settings in mind, as they enable a seamless and straightforward integration of many and various data sources to inform the dynamics. In general, we do not expect LFlows to be competitive for (hydrodynamic) forward problems compared to highly specialized classical forward solvers.
>
> Nonetheless, one key advantage is the mesh-free nature of the model without the need for representing particles explicitly (as in e.g. Smoothed Particle Hydrodynamics (SPH)). It should be noted that weakly enforcing _additional_ equations with a PDE loss might still require the use of many collocation points as in PINNs.
>
> Some additional equations can however be enforced by construction with LFlows. For incompressible fluid-flow problems, iterative solvers often require reprojections for ensuring divergence-free velocities [1]. In LFlows one could directly parameterize incompressible flows by ensuring that all time-dependent bijections have a Jacobian determinant of 1 (e.g. with shifts and rotations), resulting in divergence free velocities by construction. (The time-dependent bijections would have to be in the initial layers of $\Phi_t$, and all time-independent bijections with Jacobian determinant != 1 would have to follow in later layers.)
>
>
> [1] Guermond, Jean-Luc, Peter Minev, and Jie Shen. "An overview of projection methods for incompressible flows." Computer methods in applied mechanics and engineering 195.44-47 (2006): 6011-6045.
>
> &nbsp;
> -----
> ### Question 3
> >Does the methodology proposed in this paper have the potential to be generalized to tackle other partial differential equations (PDEs) beyond the continuity equation? If so, what challenges or modifications might be necessary when applying it to different PDEs?
>
> Our method for modeling densities is applicable to all systems of PDEs that are based on the continuity equation with a non-negative density, such as for example Fokker-Planck or Navier-Stokes like equations.
>
> As LFlows strongly rely on the push-forward of a density, we unfortunately do not see a direct way to generalize the concept to other PDEs.

---

> > ### Comment · Reviewer_1yfJ · 2023-11-21
> >
> > Thank you for the additional responses! It really helps me position this work in the literature. I would keep the current accept rating.

---

### Meta-Review · Area_Chair_egG4 · 2023-12-06

**Metareview:**

This submission introduces Lagrangian Flow Networks (LFlows), a novel approach to model fluid densities and velocities in compliance with the continuity equation. Its key innovation lies in parameterizing diffeomorphisms conditioned on time to map base densities, thereby ensuring consistent velocity expressions with density changes and obviating the need for numerical solvers or additional PDE penalties. The technique demonstrates superior predictive accuracy and computational efficiency in both 2D and 3D density modeling tasks, including a practical application in modeling bird migration using sparse radar data.

The reviewers agree that the paper stands out for its significant contribution to neural PDE methods. It introduces a novel method that marries neural networks with conditioning normalization flows, leading to a robust model that inherently satisfies key physical laws. The paper’s methodological rigor is commendable, with extensive validations conducted across various settings, including real-world datasets.

A primary concern raised by the reviewers is regarding the presentation and clarity of certain sections, particularly around the implementation of LFlows and the handling of spatially sparse data. The reviewers pointed out potential issues in the novelty of the approach, drawing parallels with existing methods like TIPF parameterization. There are also limitations associated with the use of normalizing flows and bijective layers, which might pose challenges in handling discontinuities.

**Justification For Why Not Higher Score:**

One reviewer remained concerned about the novelty of the paper, and the extensiveness of the experiments.

**Justification For Why Not Lower Score:**

The paper takes a significant stride in neural PDE methods. It provides a toolbox that will likely be helpful to future researchers.

---

### Decision · Program_Chairs · 2024-01-16

Accept (spotlight)